# Topologically-guided continuous protein crystallization controls bacterial surface layer self-assembly

Colin J. Comerci[1,2,8], Jonathan Herrmann[3,4,8], Joshua Yoon[5,2], Fatemeh Jabbarpour[3,4], Xiaofeng Zhou[6], John F. Nomellini[7], John Smit [7], Lucy Shapiro[6], Soichi Wakatsuki [3,4] & W.E. Moerner [1,2,5]

Many bacteria and most archaea possess a crystalline protein surface layer (S-layer), which surrounds their growing and topologically complicated outer surface. Constructing a macromolecular structure of this scale generally requires localized enzymatic machinery, but a regulatory framework for S-layer assembly has not been identified. By labeling, super-resolution imaging, and tracking the S-layer protein (SLP) from *C. crescentus*, we show that 2D protein self-assembly is sufficient to build and maintain the S-layer in living cells by efficient protein crystal nucleation and growth. We propose a model supported by single-molecule tracking whereby randomly secreted SLP monomers diffuse on the lipopolysaccharide (LPS) outer membrane until incorporated at the edges of growing 2D S-layer crystals. Surface topology creates crystal defects and boundaries, thereby guiding S-layer assembly. Unsupervised assembly poses challenges for therapeutics targeting S-layers. However, protein crystallization as an evolutionary driver rationalizes S-layer diversity and raises the potential for biologically inspired self-assembling macromolecular nanomaterials.

[1] Biophysics Program, Stanford University, Stanford 94305-5101 CA, USA. [2] Department of Chemistry, Stanford University, Stanford 94305-4401 CA, USA. [3] Department of Structural Biology, Stanford University, Stanford 94305-5101 CA, USA. [4] Bioscience Division, SLAC National Accelerator Laboratory, Menlo Park 94025 CA, USA. [5] Department of Applied Physics, Stanford University, Stanford 94305-4090 CA, USA. [6] Department of Developmental Biology, Stanford University, Stanford 94305-5329 CA, USA. [7] Department of Microbiology and Immunology, University of British Columbia, Vancouver V6T 1Z3 BC, Canada. [8]These authors contributed equally: Colin J. Comerci, Jonathan Herrmann. Correspondence and requests for materials should be addressed to S.W. (email: soichi.wakatsuki@stanford.edu) or to W.E.M. (email: wmoerner@stanford.edu)

Assembling a macromolecular structure on the micron scale often requires input energy and spatial coordination by enzymes and other cellular processes[1–3]. S-layers, however, exist outside the cell envelope and lack access to many cytosolic components, including ATP[4–6]. How do microbes continuously assemble a crystalline macromolecular structure on a highly curved cell surface undergoing drastic changes during normal cell growth? To answer this question, we perform time-resolved, super-resolution fluorescence imaging and single-molecule tracking (SMT) of S-layer assembly on living *C. crescentus* cells. In *C. crescentus*, the S-layer is made of a single 98 kDa SLP, RsaA, which accounts for around 30% of the cell's total protein synthesis[7]. RsaA, like other SLPs, self-assembles into crystalline sheets upon the addition of divalent calcium ($Ca^{2+}$) in vitro[8–12]. Given the many fitness-related functions ascribed to crystalline bacterial S-layers, we hypothesized that SLP self-assembly may play a role in generating the S-layer coat in vivo[4,6].

RsaA covers the cellular surface of *C. crescentus* by forming a 22 nm-repeat hexameric crystal lattice and is non-covalently anchored to an ~18 nm thick LPS outer membrane[13–17] (Fig. 1a–c). The surface topology of stalked and predivisional *C. crescentus* includes a cylindrical stalk measuring roughly 100 nm in diameter while the crescentoid cell body approaches 800 nm in width[18,19] (Fig. 1d). This large variety of curved topologies guarantees that crystal distortion and defects within the S-layer lattice structure are present and enable complete coverage of the bacterial surface[4,13,20,21]. We can use Gaussian curvature, the product of the maximum and minimum curvatures at a given point, to quantify the cellular topology[22]. Crystalline defects cluster at regions with high absolute values of Gaussian curvature such as the cell poles and division plane, while grain boundaries occur where Gaussian curvature approaches zero such as the cell body (Fig. 1d)[20,21].

Specifically labeling the S-layer in vivo has proven difficult due to the SLP's life cycle and functions, which include secretion,

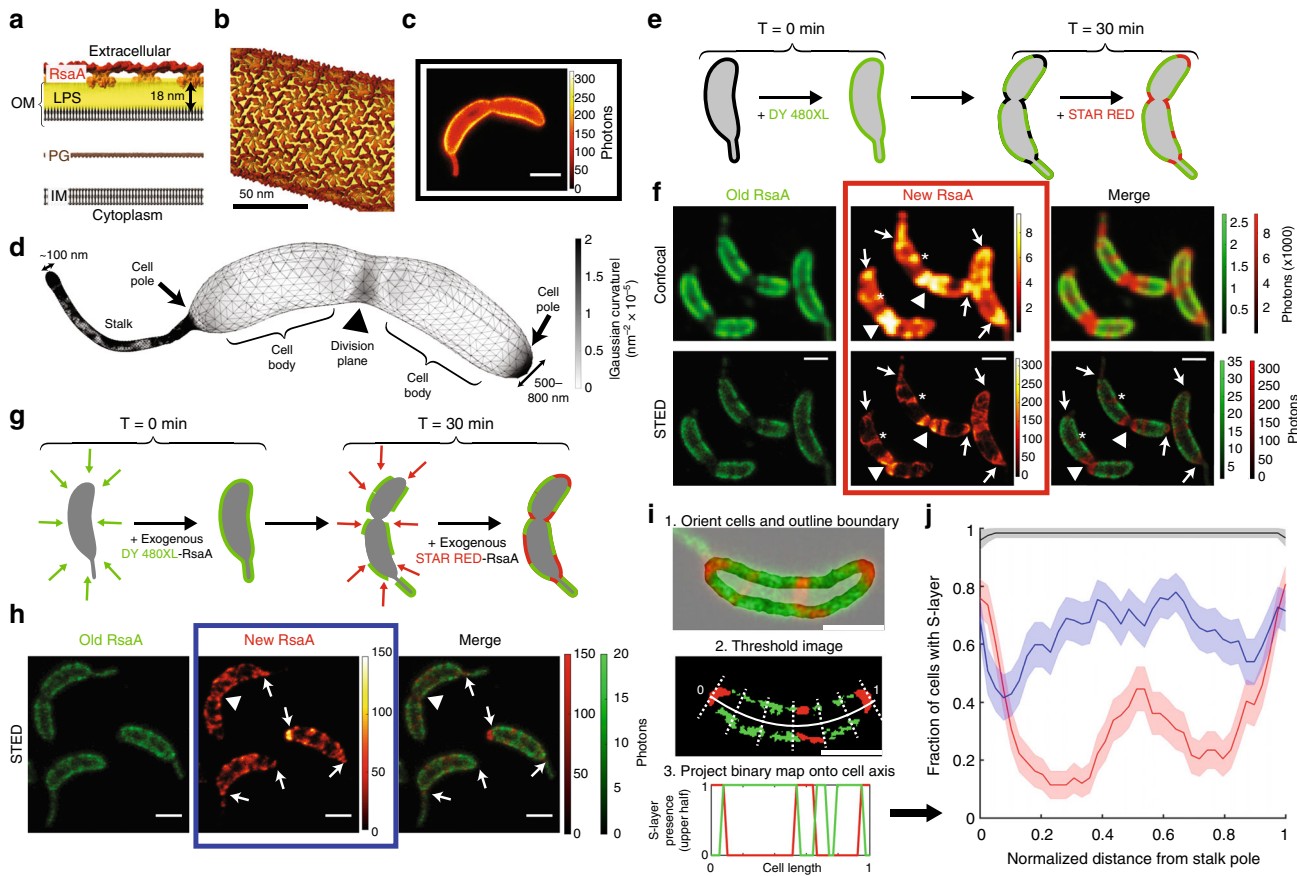

**Fig. 1** Localized S-layer assembly occurs independently of secretion. **a** Schematic of the *C. crescentus* cell envelope with the S-layer crystal lattice (red/orange) anchored to the outer membrane (OM) via an 18 nm thick LPS layer (yellow), peptidoglycan (PG), and inner membrane (IM). **b** Model of the RsaA S-layer structure (EMD-3604) applied to the surface of a 100 nm diameter cylinder. **c** STED fluorescence microscopy image of a CysRsaA cell labeled with STAR RED. **d** 3D mesh representation of predivisional *C. crescentus* topology with absolute value of Gaussian curvature projected onto the surface (gray shading). **e** Schematic of 2-color pulse-chase experiment to image the sites of S-layer assembly where fluorophores DY-480XL and STAR RED were added to CysRsaA cells 30 min apart. **f** Confocal (top) and STED images (bottom) show localized assembly of natively produced S-layer at the cell poles (arrows), division plane (triangles), and crack-like features on the cell body (asterisks). **g** Schematic of pulse-chase experiment where saturating quantities of purified, fluorescently labeled CysRsaA are added to the media of ΔRsaA cells. **h** STED images show localization of newly incorporated CysRsaA protein at the cell poles (arrows), division plane (triangle), and other locations. **i** Overview of analysis method. Top: Cells were horizontally aligned with the stalk on the left side and the cell boundary was identified (non-gray region). Middle: S-layer was identified in both channels via intensity thresholding, and the binary image is projected along the normalized cell axis. Bottom: The projected image created two binary profiles of the S-layer for the upper and lower half of the cell in each color channel (red and green lines). **j** S-layers produced natively (red, representative cell in red boxed image Fig. 1f) or by exogenous protein addition (blue, representative cell in blue boxed image Fig. 1h) show preferential incorporation at the cell poles and division plane, while control cells (black, representative cell in black boxed image Fig. 1c) show uniform labeling. Shaded regions show 95% confidence intervals for $n = 81$, 75, and 57 cells for native, exogenous, and control cells, respectively. Scale bars = 1 μm unless noted

refolding, anchoring, and crystallization[11,16,17,23,24]. Previously, electron microscopy of the *C. crescentus* S-layer was performed by inserting cysteine residues into the RsaA sequence and labeling the protein with nanogold via maleimide chemistry[13]. One such variant, henceforth referred to as CysRsaA, consists of a small tail added to the N-terminus of RsaA, which includes a single cysteine residue[13] (Supplementary Fig. 1, Methods). CysRsaA-producing cells divide normally and create a stable S-layer at a rate similar to that of WT cells, and protein can be extracted and purified as a monomeric species (Supplementary Fig. 1). By examining the dynamic behavior of S-layer assembly using specific small-molecule labels and living cells, we find that protein self-assembly alone is sufficient to create and maintain the S-layer. The topology of the cell creates localized defects and boundaries within the S-layer lattice, which guides continuous protein crystallization to specific regions of the cell surface.

## Results

**Imaging localized S-layer assembly in living cells.** Covalently modifying CysRsaA with membrane-impermeable fluorophores via maleimide chemistry is a robust, highly specific labeling scheme for RsaA and enables live-cell STimulated Emission Depletion (STED) fluorescence microscopy showing a complete S-layer (Fig. 1c; Supplementary Fig. 2). Pulse-chase STED imaging of living cells was performed using DY-480XL as the pulse fluorophore and STAR RED as the chase with a 30 min delay (Fig. 1e, f). Using only stalked and predivisional cells oriented in the same direction, we found highly localized S-layer assembly in growing cells characterized largely by new protein enrichment at both cell poles and the division plane (Fig. 1f, i, j; Supplementary Fig. 3), in agreement with observations by electron microscopy[25]. However, high-efficiency labeling coupled with super-resolution STED imaging revealed nano-scale crack-like features of new RsaA on the cell body, indicating additional sites of localized S-layer assembly (Fig. 1f). We sought to determine the factors that contribute to S-layer assembly by examining how RsaA secretion, cell wall growth, and the presence of an existing S-layer structure affect the location of S-layer assembly in living cells.

**Secretion alone does not explain localization.** To determine whether RsaA secretion is necessary to localize S-layer assembly, we added purified, fluorescently labeled CysRsaA to cultures of cells with a genomic deletion of the *rsaA* gene (ΔRsaA). Saturating quantities (600 nM) of DY-480XL labeled CysRsaA were introduced as a pulse, followed by a wash, 30 min of cell growth, and another saturating quantity of STAR RED labeled CysRsaA as chase (Fig. 1g). This experiment revealed that exogenously added CysRsaA incorporates in a manner favoring the poles and division plane (Fig. 1h). However, reconstituting S-layer assembly in this way creates more widespread, punctate fluorescence signal on the cell body (Fig. 1h, j, blue) compared to the crack-like features seen in native S-layer assembly (Fig. 1f, j, red). This effect is attributable to structural differences between the native and exogenous S-layers and is discussed below. Previous immunogold staining and electron microscopy of RsaF, the outermost component of the RsaA secretion apparatus, indicated its diffuse localization[23,26]. Therefore, the non-uniform reconstitution of S-layer assembly with purified protein suggests that a factor independent of RsaA secretion contributes to assembly localization.

**Cell wall growth alone does not explain localization.** To investigate the effect of changing cell wall surface area on S-layer assembly, we manipulated peptidoglycan (PG) insertion using the specific MreB perturbing compound, A22[27]. Under normal conditions, the cell wall grows mainly at the polar-proximal base

of the stalk and around the middle of the cell body, which eventually becomes the division plane[2]. These regions alone are insufficient to explain localized S-layer assembly due to the additional appearance of signal at the pole opposite the stalk (Fig. 1j). At 2 μg/mL, A22 delocalizes MreB but still allows some peptidoglycan addition along the cell body, leading to lemon-shaped cells that divide slowly[27] (Supplementary Fig. 4). At 25 μg/mL, A22 fully inhibits peptidoglycan insertion and surface area addition stops completely except at division planes where constriction began before drug exposure[27] (Supplementary Fig. 4). Using a pulse-chase labeling scheme after drug treatment (Fig. 2a), we found that cells treated with 2 μg/mL A22 maintain bipolar localized S-layer assembly, but the amount of new RsaA incorporated along the cell body increased (compare red and green highlighted columns in Fig. 2b, c). At 25 μg/mL A22, bipolar localized S-layer assembly is disrupted as evidenced by a decrease in signal at both poles (Fig. 2b, c, black). Native cell wall growth is asymmetric to maintain the shape of the cell and lengthen the stalk[1,2], yet S-layer assembly remains largely symmetrically localized at both poles with or without A22.

Treating cells with cephalexin prevents cell division by inhibiting peptidoglycan insertion at the division plane[28] (Supplementary Fig. 5). In cephalexin-treated cells, S-layer assembly follows untreated localization patterns, but with more crack-like features of new RsaA on the significantly elongated cell body (Fig. 2d, e). Taken together, both drug treatments show that S-layer assembly on the cell body appears to correlate with localized cell wall growth while bipolar localization is not driven by PG insertion alone.

**Assembly occurs in patches without a pre-existing S-layer.** Physically removing the S-layer from a CysRsaA cell provides a clean surface with which to observe the cell replacing its own S-layer, which we term de novo assembly. Calcium depletion (50 μM CaCl₂ instead of 500 μM CaCl₂ normally present in minimal growth medium) has been shown to cause shedding of the RsaA S-layer[29,30]. After culturing CysRsaA cells overnight in media containing 50 μM CaCl₂, supplementing the media with 500 μM CaCl₂ causes a new S-layer to be produced on the surface of *C. crescentus* (Fig. 3a). A time course of de novo S-layer assembly using only one fluorophore label at different time points displays the appearance of several S-layer patches per cell within an hour of calcium introduction (Fig. 3b, c). By 2 h, cells have produced a mostly complete S-layer, which agrees with the 2 h doubling time of *C. crescentus* (Fig. 3c; Supplementary Fig. 6). Localization analysis of de novo S-layer assembly at 30 min reveals its exclusion at the poles in sharp contrast to S-layer assembly in cells with a pre-existing S-layer, which prefers the poles and division plane (red curve, Fig. 3j).

**RsaA nucleates and crystallizes on the cell surface.** To determine how these patches grew, we performed pulse-chase imaging of de novo S-layer assembly (Fig. 3d). This experiment revealed that initial S-layer patches expand from the perimeter of each patch (Fig. 3e), consistent with nucleation and growth characteristic of in vitro SLP crystallization observed by time-resolved atomic force microscopy[10,31]. To further evaluate this mode of assembly, very low concentrations (0.5 to 10 nM) of purified, STAR RED labeled CysRsaA were added to cultures of ΔRsaA cells (OD_{600nm} = 0.2) and puncta were observed (Fig. 3f, g). The number of puncta appeared dependent on CysRsaA concentration whereas the number of molecules in each punctum did not strongly correlate with CysRsaA concentration from 0.5 to 10 nM (Fig. 3g, k). These observations are consistent with fast nucleation occurring at the measured CysRsaA concentrations and limited further growth

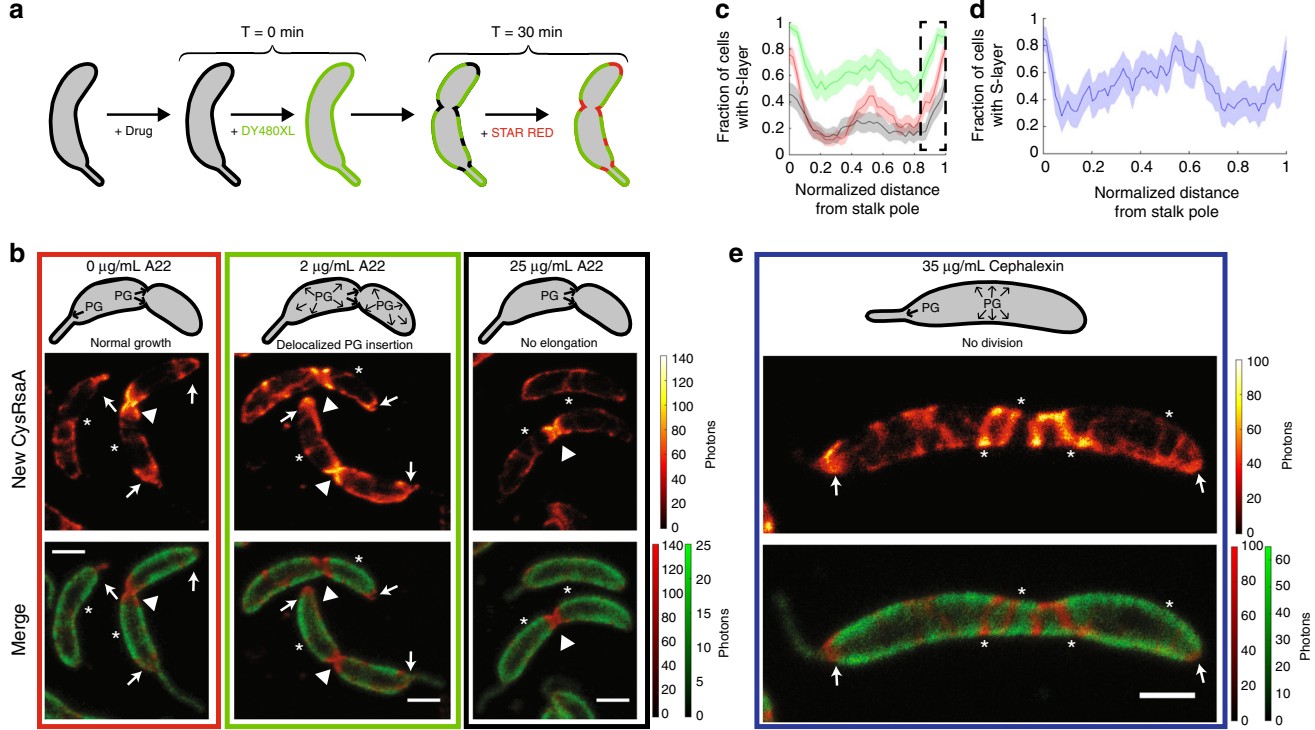

**Fig. 2** Cell wall growth is insufficient to explain polar localized S-layer assembly. **a** Schematic of pulse-chase labeling of CysRsaA cells treated with A22 (to inhibit MreB) for 1 h or cephalexin (to inhibit cell division) for 3 h. **b** Upper: Schematic showing effect of A22 treatment on PG insertion in *C. crescentus*, where untreated cells add PG primarily at the stalk pole and division plane, cells treated with 2 μg/mL A22 exhibit delocalized PG insertion along the entire cell with enrichment at the division plane, and cells treated with 25 μg/mL A22 have PG insertion exclusively at the division plane. Lower: STED images of S-layer assembly for treated cells with signal at the poles, division planes, and crack-like features marked by arrows, triangles, and asterisks, respectively. **c** Quantitation of S-layer assembly localization on cells treated with 2 μg/mL A22 (green) show similar degrees of polar localization, but enhanced localization along the cell body compared to untreated cells (red curve; reproduced from Fig. 1j). Polar localized S-layer assembly is decreased in cells treated with 25 μg/mL A22 (black), including the pole opposite the stalk (black dashed box). Shaded regions show 95% confidence intervals for $n = 81$, 54, and 63 cells for 0 μg/mL, 2 μg/mL, and 25 μg/mL A22 treatment, respectively. **d** Quantitation of S-layer assembly localization on cells treated with 35 μg/mL cephalexin show localization at both poles. Shaded region shows 95% confidence interval for $n = 25$ cells. **e** Upper: Schematic showing effect of 35 μg/mL cephalexin, which halts PG insertion during cell division. Bottom: STED image of CysRsaA cell treated with cephalexin shows localized S-layer assembly at both poles (arrows) while also adding S-layer at crack-like features (asterisks) along the cell body. Scale bars = 1 μm

of the observed puncta during the 15 min incubation. Adding 5 nM of exogenously purified and labeled DY-480XL CysRsaA followed by 10 nM of STAR RED CysRsaA revealed expansion of fluorescent puncta from the perimeter, as observed with two-color native de novo S-layer assembly (Fig. 3e, h, i). These observations are consistent with each punctum representing a nucleated RsaA protein crystal of at least 7 RsaA hexamers (Fig. 3k, inset). Large, natively grown S-layer crystals initially exclude the cell poles (Fig. 3c, j); however, introducing exogenously purified RsaA to ΔRsaA cells creates small S-layer crystals that appear more uniformly distributed along the cell axis (Fig. 3j).

**Observing RsaA diffuse and crystallize on the cell surface**. If de novo S-layer assembly occurs through nucleation and 2D crystallization of RsaA, then protein self-assembly could also be responsible for continuous S-layer growth. Since protein crystallization is concentration-dependent, we predict that once secreted, RsaA monomers should be able to diffuse while non-covalently anchored to the LPS outer membrane until incorporated into a nucleating or growing S-layer crystal. Indeed, at 1 nM exogenous CysRsaA, cells exhibit weak diffuse fluorescence suggestive of diffusing molecules (Supplementary Fig. 7). Therefore, we employed single-molecule tracking (SMT) to dynamically track the location of individual CysRsaA monomers anchored to the LPS outer membrane. ΔRsaA cells were first pre-

treated with 2.5 nM Cy3-CysRsaA to form sparse, immobile S-layer crystalline patches or seeds on the LPS outer membrane (Fig. 4a, green). Then, 1 nM AlexaFluor647-CysRsaA (Fig. 4a, red) was added to the growth media, allowing single molecules to flow into the microscope's viewing area. Once attached to the LPS outer membrane, CysRsaA molecules were imaged for several minutes at a time (Fig. 4b). CysRsaA tracks included entirely immobile molecules, entirely mobile molecules, and molecules that appear to sample mobile and immobile states during the experiment (Fig. 4b, c; Supplementary Figs 8, 9). Correlating mobility of the molecule with distance measurements from a nearby S-layer seed (Cy3 signal) reveals behavior consistent with binding at the edge of a growing S-layer crystal (Fig. 4c, d; Supplementary Fig. 9). Tracking CysRsaA anchored to the LPS outer membrane while utilizing a double helix point-spread function for 3D localization[18] allowed calculation of an apparent diffusion coefficient, $D = 0.077 \mu m^2/s$ (Fig. 4e, Methods). A comparison with the recent SMT of a similarly sized trans-membrane signaling protein anchored to the inner membrane of *C. crescentus*, CckA ($D = 0.0082 \mu m^2/s$)[32], reveals that CysRsaA diffuses an order of magnitude faster.

## Discussion
Based on these observations, we propose a model of S-layer assembly in *C. crescentus* whereby RsaA monomers are secreted

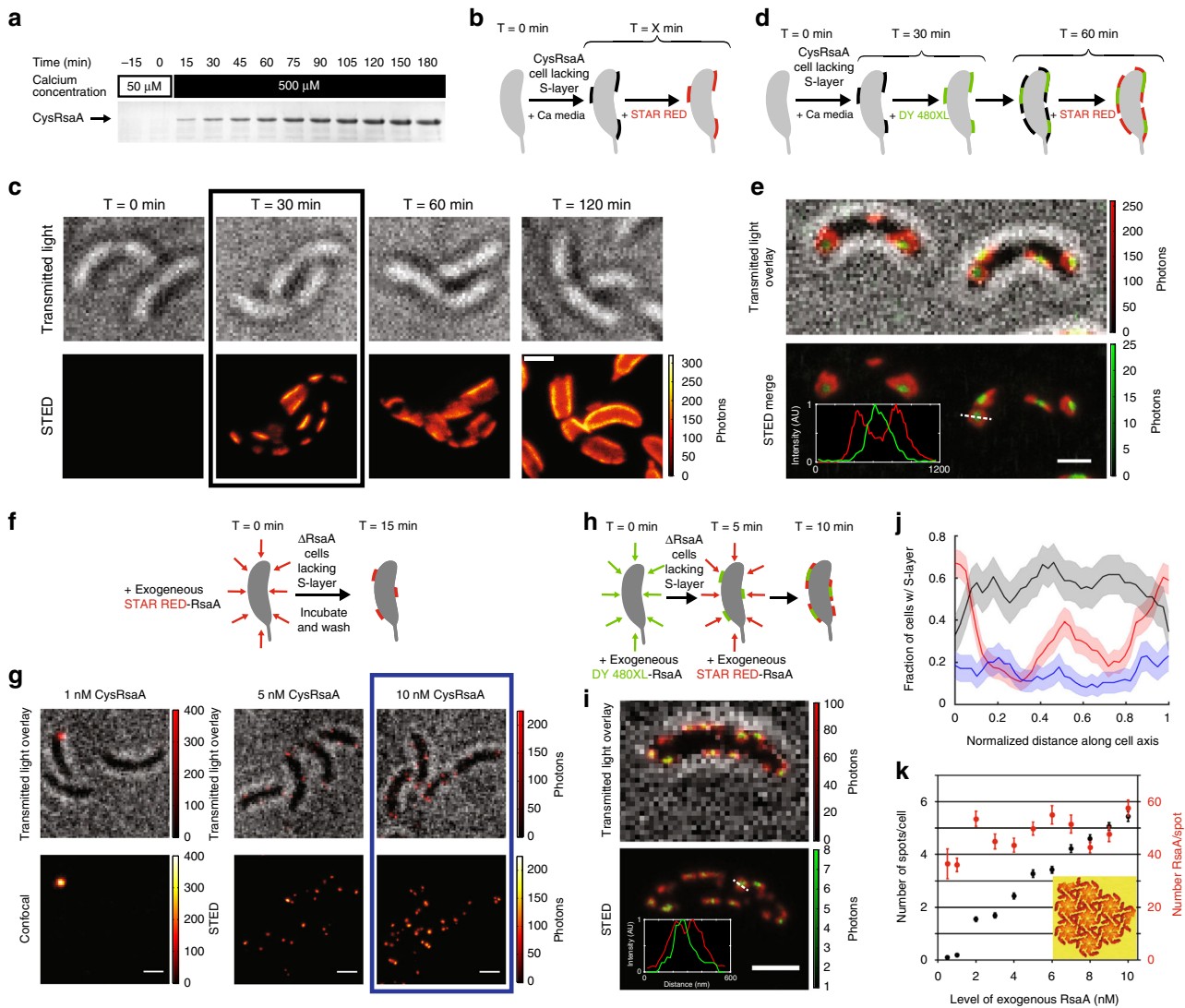

**Fig. 3** A new S-layer assembles by crystallization and location depends on crystal size. **a** Coomassie-stained SDS–PAGE of protein samples extracted from the surface of CysRsaA cells grown in low calcium M2G (50 μM CaCl₂, t < 0 min) and then switched to 500 μM CaCl₂ (0 < t < 180 mins) show de novo accumulation of the S-layer. **b** Schematic of de novo S-layer assembly experiment where CysRsaA cells are resuspended in M2G with 500 μM CaCl₂ and then their S-layer is labeled with STAR RED. **c** A STED imaging time course of STAR RED labeled CysRsaA cells shows that de novo assembly of a new S-layer occurs at discrete patches that grow larger over time. **d** Schematic of 2-color pulse-chase labeling between 30 and 60 minutes after calcium addition. **e** STED imaging reveals growth of S-layer patches from their perimeter (red signal at the edges of green patch) and is confirmed by line profiles of the dashed line (inset). **f** Schematic of experiment where low concentrations of STAR RED-labeled CysRsaA are incubated with ΔRsaA cells for 15 mins. **g** STED images show exogenous CysRsaA nucleates small puncta of S-layer. **h** Schematic of 2-color stepwise addition of 5 nM DY-480XL labeled CysRsaA followed by 10 nM STAR RED labeled CysRsaA. **i** STED imaging shows that puncta on the cell surface formed by exogenous addition of purified CysRsaA grow from their perimeter, confirmed by line profiles of the dashed line (inset). **j** Quantitation of S-layer assembly localization for native S-layer assembly (red), de novo native S-layer assembly (black), and S-layer assembly by exogenous addition of labeled CysRsaA (blue). Shaded regions show 95% confidence intervals for n = 100, 50, and 65 cells for native, de novo, and exogenous protein addition, respectively. **k** Quantitation of the number of S-layer puncta and average number of RsaA molecules per punctum on ΔRsaA cells upon addition of purified CysRsaA protein. Error bars represent SEM. A model of an RsaA crystal of about the measured size is shown (inset). Scale bars = 1 μm (**c**, **e**, **g**, **i**)

randomly and subsequently explore the LPS outer membrane by diffusion until captured and incorporated into the growing crystalline S-layer structure (Fig. 4f). While SMT of RsaA was performed on ΔRsaA cells, it is plausible that RsaA diffuses either on or within the LPS outer membrane, depending on whether an existing S-layer structure is present above the site of RsaA secretion. Protein diffusion within the LPS outer membrane in gram-negative bacteria has been observed by fluorescence recovery after photobleaching, but individual LPS molecules are immobile[33]. In *C. crescentus*, the LPS outer membrane is ~18 nm thick, while the size of a single RsaA monomer can be

approximated by its radius of gyration (Rg = 5.8 nm)[14,30]. Several crystal structures of S-layer anchoring domains from gram-positive bacteria have been elucidated, revealing multiple poly-saccharide binding sites per subunit[34,35]. Assuming that the N-terminal domain of RsaA has similar multiple carbohydrate binding surfaces, a plausible molecular basis for anchored extracellular protein diffusion is that S-layer proteins migrate along the cell surface by alternately binding and releasing the polysaccharides of neighboring LPS molecules.

Our results imply that continuous native S-layer crystallization occurs at gaps, defects, and grain boundaries within the S-layer

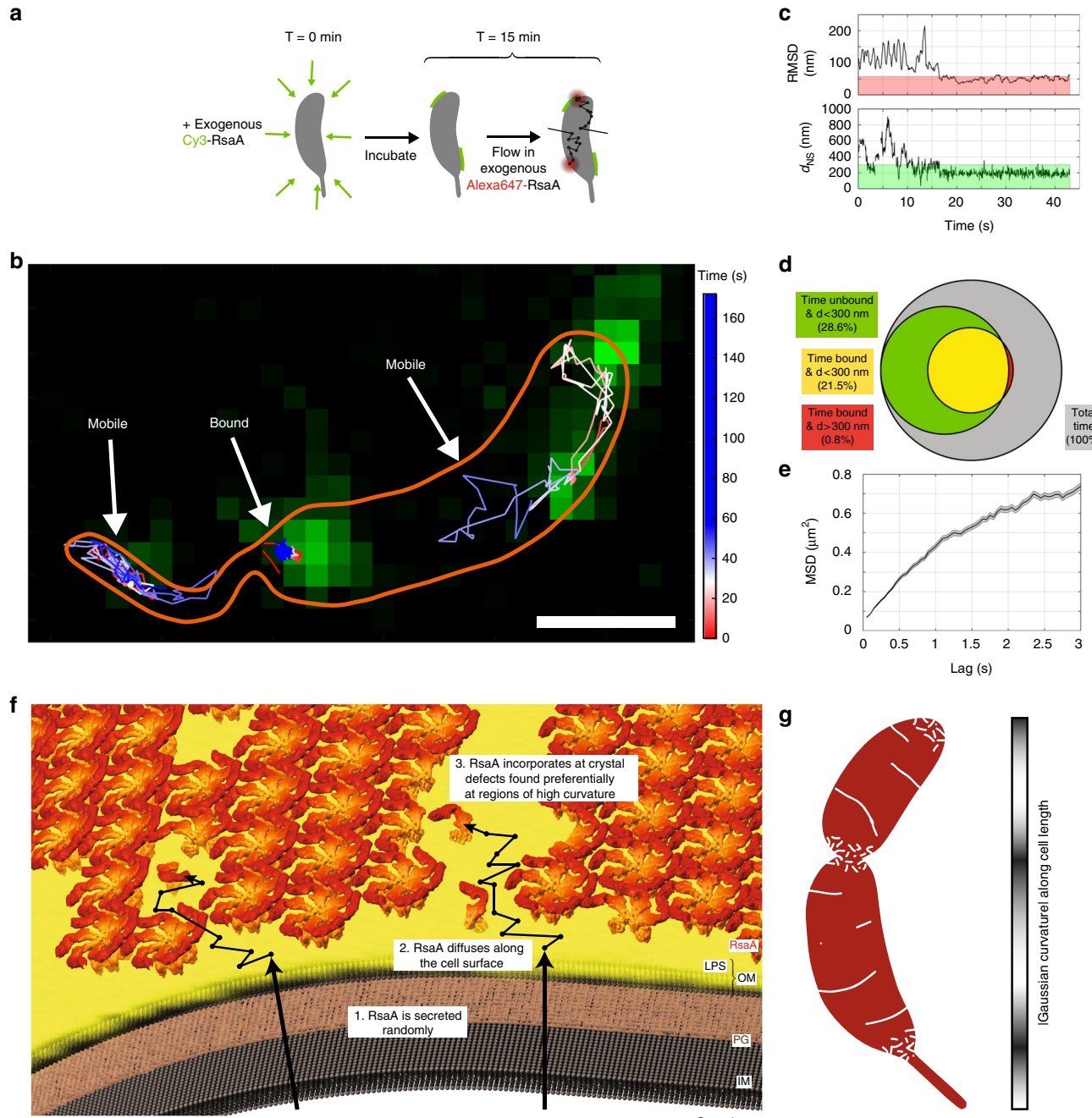

**Fig. 4** RsaA monomers diffuse on the outer surface and bind to S-layer crystals. **a** Schematic of 2D, 2-color single-molecule tracking of CysRsaA on a ΔRsaA cell. Cells were incubated with 2.5 nM Cy3-labeled CysRsaA (green) to create small fluorescent seed crystals. Single molecules of AlexaFluor647 labeled CysRsaA diffused through the agarose mounting pad and bound to the cell surface. **b** Single-molecule tracking (SMT) shows mobile and immobile single-molecule tracks. The cell boundary is sketched in orange. Scale bar = 1 μm. **c** Upper: The RMSD from a nearby seed of an example track shows binding of the molecule from ~16 to 43 s. Binding is defined as RMSD < 57.3 nm (red shaded region, see also Supplementary Fig. 8). Lower: Distance from the nearest nucleation seed ($d_{NS}$) for the same track shows the molecule binds ~200 nm from the center of a crystal patch. Green shaded region shows the 300 nm threshold used for determining crystal patch proximity. **d** A quantitative Venn diagram displaying the total time (circle indicated by gray area), and the fractions of time molecules spend near a crystal patch and unbound (green area), near a crystal patch and bound (yellow area), and bound but not near a crystal patch (red area), shows that when molecules cease diffusing, they are almost always in close proximity to an existing crystal patch. **e** MSD analysis from 3D SMT of AlexaFluor647 labeled CysRsaA molecules revealing a diffusion coefficient of D = 0.077 μm²/s. Shaded regions represent SEM. **f** A generalizable model for S-layer assembly including secretion, diffusion, and incorporation at gaps within the S-layer lattice. **g** A model for S-layer crystal boundaries/defects (white) mapped onto the *C. crescentus* surface (red), suggested by Gaussian curvature calculations (gray colorbar, right)

structure caused by localized cell wall growth or the inherent topology of the cell surface (Fig. 4g). Reconstituting S-layer assembly using the stepwise addition of exogenously purified CysRsaA yielded a non-uniform pattern of S-layer assembly, indicating a secretion-independent process (Fig. 1j). During this experiment, the long crack-like features observed on the cell body during native assembly (Fig. 1f) are replaced by more widespread, punctate fluorescence signal (Fig. 1h). Reconstituting the S-layer

with purified CysRsaA creates small, punctate S-layer crystal patches on the cell surface (Fig. 3g, Supplementary Fig. 7), which are significantly smaller than natively produced S-layer crystals (Fig. 3c, g). Therefore, the morphological difference in fluorescence signal on the cell body between native and exogenous cases could reflect the difference in molecular boundaries created by large versus small S-layer crystals. Small crystals present more boundaries for new S-layer incorporation, rationalizing the increased and less localized fluorescence signal along the cell body observed when reconstituting S-layer assembly by exogenous protein addition.

Increasing localized cell wall growth along the cell body (treatment with 2 µg/mL A22) increases S-layer assembly at that location; however, preventing asymmetric cell elongation (treatment with 25 µg/mL A22) disrupts S-layer assembly at both poles, indicating that an additional factor is coordinating this process (Fig. 2c)[2]. When the cell lacks an existing S-layer and assembles a new one using a few large crystalline patches, S-layer coverage initially excludes the poles rather than preferentially assembling there (Fig. 3j). Decreasing the size of nucleated S-layer crystals by adding exogenous RsaA to ΔRsaA cells disrupts polar exclusion (Fig. 3j), indicating that an underlying tether within the LPS is not responsible for localizing S-layer assembly.

The molecular defects and boundaries created by imposing a large crystalline lattice on a variably curved surface are sufficient to explain these observations[20,21]. Regions of non-zero Gaussian curvature values on a 3D model of the *C. crescentus* topology correlate with the polar and divisional regions of S-layer assembly observed in this study (Figs. 1d, j, 4g). At these regions, we expect small crystalline patches with more defects, allowing for more local RsaA incorporation. Along the cell body, changes in curvature are less dramatic, which produce longer grain boundaries between large crystalline patches[20,21]. The crack-like features of new RsaA we observe on the cell body further support the model that S-layer assembly patterning reflects growth at natural imperfections in the S-layer crystalline lattice (Figs 1f and 2b, e).

In *C. crescentus*, RsaA deletion and the subsequent loss of an S-layer disrupts normal cell growth, suggesting a connection between the S-layer structure and cellular fitness[30]. Given the central role of RsaA crystallization in S-layer assembly and this connection to fitness, the protein's ability to self-assemble may drive natural selection of the RsaA amino acid sequence. Remarkably small but stable RsaA protein crystals consist of only ~50 molecules (Fig. 3g, k), suggesting that fast, efficient nucleation at low concentrations may be another selectable trait supporting protein self-assembly. SLPs are exceptionally diverse in sequence, varying widely in size (40–200 kDa) and fold[4]. Functional convergence of diverse crystalline structures can be rationalized by selection driven by protein self-assembly, which can occur independently of overall fold and instead requires just a few key surface residues making symmetric, planar crystal contacts[36,37]. Similarly, diverse SLPs in archaea prefer charged (acidic or basic) amino acids to facilitate nutrient uptake through the nanoporous S-layer—a function that evolves independently of protein fold[38].

The mechanisms by which bacteria build, maintain, and evolve their S-layers are important to human health and our ability to treat and respond to bacterial pathogens such as *C. difficile*, *A. salmonicida*, and *B. anthracis*[39–41]. Additionally, S-layers have been exploited as nanomaterials in a variety of applications[4,42], including high-density organization and display of organic or inorganic molecules tethered to RsaA in particular[43,44]. This study proposes a mechanism by which bacteria can control extracellular structures without direct intracellular feedback, exploiting the biophysics of macromolecular 2D crystalline self-assembly on curved 3D surfaces. In particular, defects and natural imperfections within the S-layer lattice serve as sites of new S-layer growth. Further manipulation of this seemingly unsupervised assembly pathway may lead to treatments that target cell surface structures such as the S-layer or allow enhanced utilization of S-layers as self-assembling macromolecular nanomaterials.

## Methods

**Strains**. Three strains were used in this study and are available from the corresponding author S.W. upon request. *C. crescentus* NA1000 (ATCC 19089), which is referred to as wild type (WT) throughout the text, was used as a control for fluorescent labeling, S-layer protein production, and drug treatment response. An RsaA-negative strain of NA1000, referred to as ΔRsaA, was generated via clean genomic deletion. We amplified 802 bp upstream and 806 bp downstream of *rsaA* using primer pairs RsaAUpstreamF/R (5′-CTACGTAATACGACTCAGGCCGCG ATCAGTGCCGACGCG-3′ and 5′-ACGTTCGCTTAGGCCATGAGGATTGTCT CCCAAAAAAAATCCCACACCC-3′) and RsaADownstreamF/R (5′-TGGCCTA AGCGAACGTCTGATCCTCGCCTAG-3′ and 5′-CGGCCGAAGCTAGCGGGC CATGGTGGCCATCTGGATC-3′). The two fragments were inserted into SpeI and EcoRI-linearized pNPTS138 by Gibson assembly. The resulting plasmid was electroporated into *C. crescentus* NA1000. Deletion mutant was isolated using a two-round selection approach[45].

For fluorescence microscopy and protein purification, a cysteine mutant of RsaA, CysRsaA, was created previously[13] and obtained from the Smit Laboratory (University of British Columbia). Briefly, RsaA with a 7-residue N-terminal tail was cloned into the p4A vector and introduced to background strain JS1023 by electroporation[13,17]. JS1023 contains the repBAC operon to enable plasmid replication as well as an amber mutation within the native *rsaA* gene and a disruptive insertion within the gene for an S-layer associated protease, *sap*[17]. All liquid and plated cultures involving CysRsaA included 2 µg/mL chloramphenicol.

**S-layer protein purification**. Purified RsaA in the absence of CaCl$_2$ was previously shown to partially unfold at 28 °C[30]. Therefore, CysRsaA samples were kept cold (<4 °C) at all times unless otherwise noted. CysRsaA protein was purified similarly to previously reported methods[30,46]. CysRsaA-producing *C. crescentus* cells were grown to early stationary phase at 30 °C in PYE medium, shaking at 200 rpm. The culture was then pelleted by centrifugation and stored at −80 °C. Approximately 1 g of cell pellet was thawed on ice, re-suspended with 10 mL of ice cold 10 mM HEPES buffer pH 7.0, and centrifuged for 4 min at 18,000 rcf. This washing step was performed three times. The pellet was then separated into 10 aliquots and 600 µL of 100 mM HEPES buffer pH 2.0 was added to each aliquot. These cell suspensions were incubated on ice for 15 min and then spun for 4 min at 18,000 rcf. The supernatants were then pooled and neutralized (pH = 7) by the addition of 5 N NaOH. To remove free divalent cations and reduce cysteine side chains, 5 mM Ethylenediaminetetraacetic acid (EDTA) and 1 mM Tris(2-carboxyethyl)phosphine hydrochloride (TCEP) were added. The protein solution was then syringe filtered using a 0.22 µm PES syringe filter and 5 mL were injected onto a Highload Superdex200 16/600 size exclusion column (GE Healthcare). During size exclusion chromatography, the running buffer consisted of 50 mM Tris/HCl pH 8.0 and 150 mM NaCl. Monomeric CysRsaA consistently eluted at ~0.55 column volumes (Supplementary Fig. 1). From 1 g of pelleted cells, we consistently purified at least 1 mg of monomeric CysRsaA protein. Purity was assessed by SDS–PAGE (Supplementary Fig. 1).

**Time-resolved Blot**. WT or CysRsaA-producing *C. crescentus* cells were grown in modified M2G medium containing 50 µM CaCl$_2$ (normally 500 µM) to log phase (OD$_{600nm}$ = 0.5). At t = 0 min, CaCl$_2$ was adjusted to 500 µM using a sterile 1 M stock. The culture was then incubated at 30 °C, shaking at 180 rpm with 0.5 mL aliquots removed every 15 minutes. Aliquots were immediately spun down and snap frozen in LN$_2$. Soluble RsaA extraction was performed as above (without chromatography) and analyzed by SDS–PAGE.

**Fluorescent labeling of the S-layer protein**. CysRsaA protein was fluorescently labeled by covalent maleimide chemistry both in vivo and in vitro. For in vivo experiments, 1 µM STAR RED (Abberior) or DY-480XL (Dyomics) Cys-reactive fluorescent label was introduced to 1 mL of log-phase (0.1 < OD$_{600nm}$ < 1.0) CysRsaA cells in minimal medium (M2G) at 30 °C. After 15 minutes, cells were washed with 1 mL of M2G once if another fluorophore was to be added next or twice if the next step was imaging. Complete labeling was evidenced by highly spatially complementary fluorescent images in pulse-chase labeled cells (Fig. 1b).

For in vitro experiments, purified CysRsaA was buffer exchanged into 50 mM HEPES pH 7.0 and 150 mM NaCl using a 30 kDa MWCO centrifugal concentrator (Sartorius). Overnight labeling of CysRsaA protein (>20 µM) was performed on ice with the addition of 1 mM TCEP and at least 5-fold stoichiometric excess of maleimide-derivatized, cell-impermeable fluorescent dyes, STAR RED and DY-480XL, which are well-suited for confocal and STED imaging. Cell-impermeable Sulfo-Cy3 (Lumiprobe) was used as a diffraction-limited indicator for S-layer seed crystals. AlexaFluor647 (ThermoFisher) was used to track single CysRsaA molecules since SMT requires a particularly bright and photostable fluorophore.

The next day, three successive 1:30 dilutions were performed to remove unbound dye molecules using a 30 kDa MWCO centrifugal concentrator (Sartorius) and buffer containing 50 mM Tris/HCl pH 8.0 and 150 mM NaCl. Absorbance measurements at 280 nm and the known absorbance peak for each fluorophore determined labeling efficiency, which varied from 50 to 90%.

**Growth curves.** For growth curve analysis of WT and ΔRsaA *C. crescentus* strains, 10 μL of mid-log phase cultures ($OD_{600nm} = 0.5$) were added to 90 μL of M2G containing varying amounts of A22 (Cayman Chemical) or cephalexin (Frontier Scientific) in a sterile, black-walled, 96-well transparent plate (Corning). While incubating the plate at 29 °C and shaking at 600 rpm between readings, $OD_{600nm}$ was measured every 5 min for up to 23 h using an Infinite M1000 microplate reader (Tecan).

**Confocal and STED microscopy.** Images were acquired on a bespoke 2-color fast scanning STED microscope[22]. Briefly, the pulsed 750 nm depletion beam is provided by a titanium-sapphire mode-locked oscillator (Mira 900D, Coherent) running at 80 MHz, providing an average power of 120–130 $MW/cm^2$ at the sample plane. A vortex phase plate imparts the STED donut shape (RPC Photonics). 530 nm and 635 nm pulsed diode lasers are used for excitation (LDH-P-FA-530B & LDH-P-C-635B, PicoQuant), providing an average power of 40–60 $kW/cm^2$ and 50–80 $kW/cm^2$, respectively. The laser beams are focused and fluorescence is collected through an oil immersion objective (Plan Fluor 100 × /1.3 NA, Nikon). A 7.5 kHz resonant mirror (Electro-Optical) scans the beams along the fast axis, while the slow axis is scanned using a piezo stage (PD1375, Mad City Labs). Fluorescence is collected through a ∼0.7 Airy unit (AU) and 0.8 AU pinhole (red and green channels, respectively), spectrally filtered from 650 to 710 nm or 550 to 615 nm, and detected on a Si APD detector (SPCM-ARQH-13, Perkin Elmer). Microscope control and image acquisition arose from a custom LabView program running on an FPGA (PCIe-7842R, National Instruments). Confocal images are taken using no depletion laser, a pixel size of 100 nm, and an average pixel dwell time of ∼0.25 ms/pixel. STED images have a pixel size of 20 nm, and an average pixel dwell time of 0.1 ms/pixel for the red channel. Images in the green channel are the sum of two frames, smoothed with a σ = 0.9 pixel Gaussian filter and an average pixel dwell time of 30μs/pixel/frame. STED images have a FWHM resolution of ∼60 nm and ∼80 nm in the red and green channels, respectively[22,47].

**Binary cell profile analysis.** S-layer cell profiles were analyzed using a custom MATLAB algorithm (Supplementary Figure 3). STED images were aligned to the gradient of the transmitted light image via cross-correlation to correct for sample drift during scanning. Well-separated cells were selected by hand and aligned to be horizontal with the stalk on the left side using a radon transform (except Fig. 3j, where stalks cannot easily be identified on cells for de novo assembly and exogenous addition experiments). A cell axis was determined by fitting a second order polynomial to the maximum intensity transmitted light pixels. For the cephalexin treated cells, a smoothing spline was used instead of a second order polynomial due to the longer cell length. Cells were classified as swarmer or stalked, and swarmer cells were omitted from analysis. For each cell, a 200 nm wide cell outline was defined. For conditions with a mostly complete S-layer (Figs. 1d and 2c), first the outer cell boundary was determined by thresholding a normalized sum of the two-color channels, and then this outer boundary was eroded to form the 200 nm cell outline. For analysis including conditions with an incomplete S-layer (Fig. 3e), the center of the cell outline was determined by finding the maximum gradient of the transmitted light image around the cell axis, followed by expansion to a 200 nm wide cell outline. Then, the presence of S-layer was determined by applying a binary intensity threshold to the STED images smoothed using a Gaussian filter with σ = 0.9 pixels (18 nm). An upper and lower binary cell profile were determined by projecting each pixel onto the cell axis and binning into 40 equi-length bins, where a single pixel identified as positive for S-layer in a bin makes the entire bin positive. The fraction of cells with S-layer at each position along the normalized cell axis can then be computed (combining both upper and lower halves), yielding a binomial observation for the probability of finding S-layer at a given position. 95% confidence intervals were determined using the Wilson score interval. All imaging experiments except for Figs 2e and 3e, i were performed on cell populations from at least two independently created samples.

**Puncta quantification.** Counting the number of crystal puncta per cell upon exogenous addition of RsaA was performed by identifying spots with signal greater than 10 standard deviations above background. Quantitation of the number of RsaA monomers per punctum was performed using a custom MATLAB algorithm. In-focus puncta were fit using non-linear least squares to an asymmetric Gaussian. These fits were used to determine the number of photons per punctum. The number of photons per RsaA monomer was determined by imaging single molecules of RsaA-STAR RED in vitro on a poly-L-lysine coated coverslip. By calculating the photons per molecule of both STED and confocal images of the in vitro sample, a photobleaching correction factor was determined by comparing the number of molecules identified in both confocal and STED images of identical fields-of-view. The final number of RsaA monomers per punctum was corrected for photobleaching, as well as the in vitro labeling efficiency of the exogenously added

RsaA, as determined by absorbance at 280 nm and the absorbance of STAR RED at 640 nm.

**2D and 3D single-particle tracking microscopy.** Images were acquired on a custom-built 2-color inverted microscope (Olympus, IX71) for imaging Cy3 and AlexaFluor647 (Supplementary Figure 10). Labeled biological samples were mounted on a 2D micrometer stage and in contact with an oil-immersion objective (Olympus, ×100, 1.4 NA, UPLANSAPO). Shutters were used in a sequential, interleaved fashion, where the sample was first exposed to the 641 nm laser (Coherent Cube, 100 mW) for 1.8 s at an intensity of 88.5 $W/cm^2$, then the 514 nm laser (Coherent Sapphire, 100 mW) for 0.2 s at an intensity of 1.6 $W/cm^2$. Emission is collected with the objective, passes through a dichroic filter (Semrock, FF425/532/656-Di01), and another dichroic filter (594LP), which allows us to spectrally separate the emission. Alexa647 emission is detected on one EMCCD camera (Andor, DU-897U-CS0-#BV) with two emission filters (Chroma, 680–60; Chroma, 655LP) and Cy3 is detected on a separate EMCCD camera (Andor, DU-897U-CS0-#BV) with an emission filter (Semrock, 578–105), both recorded at a framerate of 20 frames/s and an electron-multiplying gain of 200. During the first ∼10 s of acquisition, the intensity of the 641 nm laser is briefly increased to a higher intensity to allow the fluorescent dyes to bleach down approximately to the single-molecule regime before reducing the intensity for optimal tracking. Spatial correlation between the two cameras is determined by detecting Tetraspeck beads (Invitrogen), which appear in both channels. 3D tracking was performed with a double-helix phase mask placed at the Fourier plane in the detection side of our microscope. Imaging conditions are similar to our 2D single-particle tracking experiments, except the 641 nm laser was used for detecting labeled RsaA molecules with an intensity of 885 $W/cm^2$. All image acquisition was performed through software made by Andor. Pixel size is 163 nm. All tracks shown (Fig. 4b and Supplementary Figure 9) are down-sampled 5X for clarity.

**3D Calibration with Fiducials.** In 3D single-molecule tracking, the double helix point spread function (DH-PSF) allowed extraction of the *xyz* position of individual emitters in the field of view[48]. Spatial calibrations utilized fluorescent beads (FluoSpheres 0.2 μm, crimson fluorescent (625/645)), spin-coated from 1% poly-vinyl alcohol onto a glass coverslip. Using a piezo-electric z-motion stage, calibrations were acquired over a 3 μm range along the z axis with a 50 nm step-size with 30 frames measured at each z-height. This calibration step produces template images of the DH-PSF, which are used for the identification of single-molecule signals during post-processing of the raw data. All imaging was performed at 25 °C. Using the *easyDHPSF* MATLAB program[49], a z-axis calibration over a 3 μm range is obtained via a 2D Double-Gaussian fit, which provides us with *xy* positions, width, amplitudes, and offset levels of each lobe of the fluorescent bead.

**Mobility analysis of 2D RsaA tracking data.** Images of single molecules and beads were analyzed using custom-built MATLAB code. Within one *C. crescentus* cell, a 2D symmetric Gaussian fit is applied to a single labeled RsaA molecule of interest, which provides an estimate of its *xy* position at that point in time. Linking together the trajectory of the same molecule over time generates tracks. In order to determine whether the detected molecules were bound or not, we first calculated the root mean squared deviation (RMSD) from the mean position over a 1 s sliding window (20 frames). Bound molecules will have a relatively low RMSD (near the localization precision) compared to a molecule freely diffusing within the cell. By performing this for every track (9 tracks) within the same cell, we were able to generate a histogram of all RMSD values (6625 RMSD values), which shows a clear peak with a tail (Supplementary Figure 8), where the left-most peak arises from the localization precision error for bound molecules. The Gaussian fit of the lowest RMSD population was utilized to determine a binding threshold (RMSD < 57.3 nm) corresponding to 2σ above the mean. Molecules were classified as bound if RMSD < Threshold or unbound if RMSD ≥ Threshold).

**2-Color analysis of 2D RsaA/nucleation site tracking data.** The *xy* location of each nucleation site, labeled with Cy3, was separately determined by fitting a 2D symmetric Gaussian. The locations of these nucleation sites was found to be stationary over the ∼15 min imaging period. Using the 2D RsaA trajectories analyzed earlier, we calculated the $d_{NS}$, which is defined as the distance between the RsaA molecule and the nearest nucleation site. By calculating both the RMSD and the $d_{NS}$, we can categorize each frame as the following: bound only, bound and close to a nucleation site, or close to a nucleation site only (Supplementary Figure 9).

**3D Mean Square Displacement (MSD) Analysis.** Images of RsaA imaged in 3D were analyzed using custom-built MATLAB code for analyzing DH-PSF data. A 2D Double-Gaussian fit was applied to each emitter in the field-of-view, which provides us with x, y, and θ information from the tilt of the two lobes of the double-helix. We use the calibration obtained earlier to convert our estimates to *xyz* values. For each individual track, the MSD is computed over a series of time lags starting from 50 ms. We then pool the data over all 30 trajectories to obtain a 3D MSD plot. The diffusion coefficient is extracted by fitting the following equation to the first 4

time lags:

$$\mathrm{MSD}_{3D} = 6D\left(\tau - \left(\frac{\tau_E}{3}\right)\right) + 2(s_1^2 + s_2^2 + s_3^2) \quad (1)$$

where D is the diffusion coefficient, $\tau$ is the time lag, $\tau_E$ is the exposure time of the camera (50 ms), and $s_i$ is the localization error in the $i^{th}$ dimension ($s_{x,y,z}$ = 93 nm, 93 nm, 91 nm).

**Gaussian curvature analysis**. Point cloud localization data of positions on the *C. crescentus* surface were obtained previously by a covalently surface-attached fluorophore[18]. A surface mesh was extracted with the Poisson Surface Reconstruction algorithm, available through MeshLab, an open source software package[50]. Gaussian curvature information can be extracted from the computed mesh[22].

**Reporting summary**. Further information on research design is available in the Nature Research Reporting Summary linked to this article.

## Data availability

Data supporting the findings of this manuscript are available from the corresponding authors upon reasonable request. A reporting summary for this Article is available as a Supplementary Information file. The code used in this study is either open access[49,50] or is described above and available upon reasonable request. Matlab analysis was performed using version 9.0.0.341360. The source data underlying Figs 1j, 2c, d, 3a, j, k, 4c-e and Supplementary Figs 1b, c, 3c, 4a, b, 5a, b, 6c, 8, 9 are provided as a Source Data file.

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

## Acknowledgements

This work was supported in part by the National Institute of General Medical Sciences: Grants No. R01-GM086196 to W.E.M. and L.S., R35-GM118067 to W.E.M., and R35-GM118071 to L.S., and in part by the US Department of Energy, Laboratory Directed Research and Development under contract No. DE-AC02–76SF00515 and Biology and Environmental Research to S.W. L.S. is a Chan Zuckerberg Biohub Investigator. J.S. was supported by the Natural Sciences and Engineering Research Council of Canada Discovery Program (Grant No. RGPIN 36574–11). C.J.C., J.H., and J.Y. were supported in part by the National Science Foundation Graduate Research Fellowship Program (NSF-GRFP). J.H. was supported in part by the US Department of Energy Office of Science Graduate Student Research Program (DOE-SCGSR). The authors thank Greg Stewart/SLAC for graphic design support.

## Author contributions

Conceptualization: C.J.C., J.H., L.S., S.W., and W.E.M. Investigation: C.J.C., J.H., J.Y., and F.J. Formal analysis: C.J.C., J.H., J.Y., X.Z., and W.E.M. Resources: X.Z., J.N., J.S., and L.S. Writing—original draft: C.J.C. and J.H. Writing—review and editing: All authors. Supervision and funding acquisition: J.S., L.S., S.W., and W.E.M.

## Additional information

**Competing interests:** The authors declare no competing interests.

