## [Peer Review File · Nature Communications]

Reviewers' Comments:

Reviewer #1:

Remarks to the Author:

Comerci et al. present a very nice study on the assembly of the RsaA S-layer during cell growth of the model bacterium *Caulobacter crescentus*. Using STED-based super-resolution microscopy and a series of pulse chase experiments using fluorescently labelled subunits, the authors follow the site of assembly of S-layer subunits that are either secreted de novo or are added exogenously. This is done both in the background of a pre-existing S-layer as well as on a *rsaA* null mutant or on cells where the S-layer was chemically removed.

Under pristine conditions – i.e. no prior S-layer, the sites of S-layer assembly are found to be patchy and stochastic, following spontaneous nucleation and 2D growth by incorporation and crystallisation of subunits laterally diffusing on the cell surface. Spontaneous nucleation and growth preferentially here occur at sites of minimal curvature.

In growing cells with pre-existing S-layer, however, assembly is highly localised, and predominantly occurs at the cell poles and the division plane, regions of higher Gaussian curvature that flank the zone of lateral elongation in *C. crescentus*. Additionally, new subunits can be filled in at crack-like zones along the cell body, presumably corresponding to regions with localised topological defects in the S-layers of growing cells.

The mechanism by which bacteria build and maintain their cell envelopes during cell growth and cell division is of great interest to a broad audience. The loss of a coordinated cell envelope expansion leads to defects in cell integrity, reduced cell growth or even cell death. Indications are that this vulnerability may include S-layer integrity, a point that is of particular interest in S-layer carrying pathogens. This study provides unprecedented insights into the mechanism of S-layer assembly in function of cell growth of the model organism *C. crescentus*. The experiments are carefully executed and make use of state of the art STED imaging and image processing.

There is one point that warrants some further exploration or discussion prior to publication:

In Fig1j, and described in lines 92 – 98, the author show the site of assembly for de novo secreted RsaA is highly localised at the cell poles and division plane. Based on the previously reported diffuse localisation of the RsaEDF Type 1 secretion machinery, the other authors conclude that there is a lack of coordination between the point of secretion and the site of incorporation in normal growing cells. This is possibly contradicted by the experiments using exogenously added RsaA, where the plot in Fig1j shows a much higher frequency of subunit incorporation along the cell body, very similar to that at the growth poles or division plane.

This needs some more investigation or more careful discussion. Is there a difference in crack generation or crack filling in the experiments monitoring endogenously secreted versus exogenously added RsaA? The author could use AFM to look into the S-layer integrity of either regimes.

Reviewer #2:

Remarks to the Author:

Summary

Comerci, C., *et al.* use super-resolution microscopy techniques to follow the diffusion and assembly of the *Caulobacter crescentus* S-Layer protein on the bacterial outer membrane. To date, self-assembly of the S-layer protein, RsaA, have been primarily studied by molecular biology or *in vitro* studies. RsaA along with other S-layer proteins from different bacterial species have been extensively studied using high resolution imaging techniques such as AFM and cryoEM. This manuscript offers a complete study of the secretion and self-assembly mechanism of RsaA on a bacterial cell surface. Using STED microscopy and single molecule tracking, the authors

learn that the topology of this crescent shaped bacterial influences the self-assembly of RsaA. Sub-diffraction microscopy techniques were key to imaging these assembly dynamics and elucidating the mechanism. The experiments and controls in this manuscript are well conceived. The scientific conclusions that are drawn are sound and reveal previously unknown aspects of the S-layer self-assembly process *in vivo*. I invite the authors to revise the manuscript by addressing the following comments.

Specific Comments

- The p4A plasmid system in the *C. crescentus* strain JS1023 was used in this manuscript and shown to express the same levels of RsaA as well as the CysRsaA. Recent work on RsaA expression and display has reported that the p4B plasmid may result in over-expression of RsaA as well as plasmid instability (Charrier et al 2019, DOI: 10.1021/acssynbio.8b00448). Does the expression rate of RsaA affect secretion and assembly of this S-Layer on the surface? Would stably integrating the CysRsaA gene in the genome with the native promoter affect the dynamics of assembly?
- On page 5, line 90 the manuscript states that “Saturating quantities (600 nM) of DY-480XL-labeled CysRsaA were introduced...” How was this RsaA protein concentration determined to be saturating? Was the saturation determined relative to the total surface area of all the cells in solution?
- On page 5, line 93 the manuscript states that exogenously added DY-480XL-labeled CysRsaA incorporates preferentially at the poles and division plane in addition to increased incorporation along the body. However, Figure 1j indicates that the fraction of labeled CysRsaA incorporation along the body is similar to that at the poles. Later in Figure 3f,g,j you again show that exogenously added labeled CysRsaA results in even incorporation of the protein along the cell. The explanation of these data sets should be more consistent and specific to avoid ambiguity.
- On page 8, the paragraph beginning on line 8 states that calcium depletion results in shedding of the RsaA S-layer. The referenced article (Ref. 30) indeed shows RsaA sheds when first grown in M2G media with high calcium and then shifted to low (8 μ M) calcium. However, this shift to a low calcium media also results in increased aggregation of RsaA. At 50 μ M calcium, presumably the RsaA is being excreted and should attach to the cell LPS even though it is not crystalline. The authors should show whether aggregates are present in these cultures (Figure 3a shows only crystalline RsaA) and address whether aggregates may affect the kinetics of RsaA nucleation and assembly as observed in Figure 3c and 3e.
- In Figure 3g,k, the authors add exogenous STAR Red-CysRsaA and measure puncta that form on the cell surface. These puncta are determined to have ~50 monomers of RsaA using photon counting. Is the invariance in size of the puncta with increasing RsaA concentrations due to the incubation time? The manuscript mentions in the Materials and Methods section that incubation is typically 15 minutes. If nucleation occurs faster than the incubation time, the puncta reflect nucleation followed by some growth. Moreover, the number of spots/cell appear to increase in an exponential fashion with the concentration of CysRsaA. Fitting a simple nucleation model may give a minimal nucleus size for RsaA on the cell surface.
- The authors show S-layer growth data in Figure 3e,i. Is it possible to estimate the observed growth of RsaA S-layers on the cell surfaces? Can an estimate of the growth rate be determined and compared to crystal growth rates obtained for other S-layers?
- In the SMT experiments, the manuscript does not explain why two different dyes other than DY-480XL and Star Red were used? Were the AlexaFluor dyes more stable? Are the AlexaFluor dyes also cell impermeable?
- In Supplemental Figure 9, it is not clear how the molecules in each panel were classified. Which

molecule was classified to be bound, bound and close to a nucleation site, or close to a nucleation site? In addition, a color bar indicating the time for the Upper panels in each figure would be helpful.

- On page 10, line 157 SMT is introduced without defining the acronym "Single Molecule Tracking."

Reviewer #3:

Remarks to the Author:

The manuscript by Comerci, Herrmann, et al. investigates the assembly of the surface layer in *C. crescentus* bacteria using STED microscopy and single-molecule tracking. Based on their observations, they propose a model of S-layer assembly whereby RsaA monomers are secreted diffusely on the outer membrane and diffuse until incorporated into S-layer crystals where they contribute to the 2D crystal growth.

The manuscript is well-written, experiments are conducted carefully using a range of cutting-edge imaging methods and the results are convincing. The findings and model are of significant interest in that they shed light onto this fundamental mechanism of self-assembly which might also have an impact in future drug development. I therefore support publication in Nature Communications.

I have only a few minor comments:

- (1) In Fig. 1, the native RsaA seems to form a smoother distribution (Fig. 1f) than the exogenously added RsaA which appears in more puncta-like structures (Fig. 1h). The images in Supp Fig.3 show a similar pattern. It would be interesting to hear a possible explanation for this phenomenon from the authors, if it should turn out to be more than just a coincidence from the selected images.
- (2) The authors did a good job analyzing and presenting the results of the STED images (Supp Fig. 3). I understand the focus of this paper is about the new RsaA signals. However, since the authors already analyzed the old RsaA signals, it would be interesting to show the curves of old signals as well (Fig. 1j).
- (3) The statement in lines 112-114 "If PG insertion were the only driver of S-layer assembly, disruption should have occurred only at the stalked pole, rather than both." is hard to follow without additional explanations.
- (4) It would be helpful to report the approximate resolution of the STED images in both channels to put the observations in context of the size of individual RsaA molecules, etc.

Reviewer #4:

Remarks to the Author:

In this work, the authors used super-resolution fluorescence microscopy and a pulse-chase scheme to propose that localised S-layer assembly in *C. crescentus* is driven by areas of high Gaussian curvature, and that S-layer growth relies on self-assembly of RsaA molecules. The authors propose a mechanism for S-layer assembly that involves diffusion of secreted RsaA molecules into the growing S-layer and incorporation at compromised sites. This work builds on previous in vitro studies showing RsaA self-assembly, and would be of interest to people in their specific field. Given that the S-layer integrity is important for cell viability do I find it curious and maybe even unlikely that its assembly would be purely diffusion driven. Overall, the paper is well-written for the most part, experiments are sound, and the data carefully interpreted. However, regrettably I don't see this study attract attention from the large and diverse readership of Nature Communications.

No major criticisms, but some suggestions that could be considered to strengthen their claims:

- Line 121 and Figure 2: The authors suggest that, in A22-treated cells, increased incorporation of CysRsaA along the cell body is due to delocalized PG insertion. Apart from the graphical depiction in Fig. 2b, the authors do not show the effect of A22 on sites of PG insertion. It would be more convincing if sites of PG insertion could be labelled (eg. using FDAAs) and correlated with sites of S-layer assembly.
- Concerning Figure 3a (line 125 onwards), it is not immediately clear that the media is exchanged, the word "reintroducing" can be changed to avoid confusion.
- Lines 95-98: The authors claim that diffuse localisation of the RsaF component of the RsaA secretion machinery and incorporation of RsaA to the cell poles and division plane suggests that S-layer assembly is independent of RsaA secretion. Perhaps the authors could correlate RsaF localisation with S-layer assembly in the same cell. However, the authors also infer diffusion of RsaA molecules from particle tracking experiments (Fig. 4b), which also suggests that S-layer assembly is independent of RsaA secretion, so this is not so critical to show.
- In its current form does this manuscript read like it has been written up as a short communication. The figures are therefore sometimes very dense and is consequently quite hard to follow. Perhaps would it be advisable to split the figure to make them more reader friendly?

Response to Reviewers

“Continuous, Topologically Guided Protein Crystallization Controls Bacterial Surface Layer Self-Assembly”

Comerci and Herrmann *et al.*

Reviewer comments are reproduced in plain font.

Author responses are bolded.

Changes to the manuscript are indicated by yellow highlight.

Reviewer #1 Remarks

*Comerci et al. present a very nice study on the assembly of the RsaA S-layer during cell growth of the model bacterium *Caulobacter crescentus*. Using STED-based super-resolution microscopy and a series of pulse chase experiments using fluorescently labelled subunits, the authors follow the site of assembly of S-layer subunits that are either secreted de novo or are added exogenously. This is done both in the background of a pre-existing S-layer as well as on a *rsaA* null mutant or on cells where the S-layer was chemically removed.*

Under pristine conditions – i.e. no prior S-layer, the sites of S-layer assembly are found to be patchy and stochastic, following spontaneous nucleation and 2D growth by incorporation and crystallisation of subunits laterally diffusing on the cell surface. Spontaneous nucleation and growth preferentially here occur at sites of minimal curvature.

*In growing cells with pre-existing S-layer, however, assembly is highly localised, and predominantly occurs at the cell poles and the division plane, regions of higher Gaussian curvature that flank the zone of lateral elongation in *C. crescentus*. Additionally, new subunits can be filled in at crack-like zones along the cell body, presumably corresponding to regions with localised topological defects in the S-layers of growing cells.*

*The mechanism by which bacteria build and maintain their cell envelopes during cell growth and cell division is of great interest to a broad audience. The loss of a coordinated cell envelope expansion leads to defects in cell integrity, reduced cell growth or even cell death. Indications are that this vulnerability may include S-layer integrity, a point that is of particular interest in S-layer carrying pathogens. This study provides unprecedented insights into the mechanism of S-layer assembly in function of cell growth of the model organism *C. crescentus*. The experiments are carefully executed and make use of state of the art STED imaging and image processing.*

The authors thank the reviewer for a careful and complete consideration of the manuscript.

Reviewer #1 Specific Comments

There is one point that warrants some further exploration or discussion prior to publication:

In Fig1j, and described in lines 92 – 98, the author show the site of assembly for de novo secreted RsaA is highly localised at the cell poles and division plane. Based on the previously reported diffuse localisation of the RsaEDF Type 1 secretion machinery, the other authors conclude that there is a lack of coordination between the point of secretion and the site of incorporation in normal growing cells. This is possibly contradicted by the experiments using exogenously added RsaA, where the plot in Fig1j shows a much higher frequency of subunit incorporation along the cell body, very similar to that at the growth poles or division plane.

This needs some more investigation or more careful discussion. Is there a difference in crack generation or crack filling in the experiments monitoring endogenously secreted versus exogenously added RsaA? The author could use AFM to look into the S-layer integrity of either regimes.

The authors agree that during exogenous addition, S-layer assembly along the cell body (shown in Fig 1j, blue) increases when compared to native S-layer assembly (red) and that the amount of signal at mid-cell is similar to the amount of signal at the poles. However, the localization of exogenously added S-layer is non-uniform and shows dips in signal at the non-polar regions (blue and red). Therefore, in the absence of secretion, S-layer incorporation still occurs preferentially at the poles and mid-cell. This result indicates that another variable not attributable to secretion exerts some amount of localization control over S-layer assembly. The manuscript text has been modified to clarify these findings.

Results: Lines 92-103

“This experiment revealed that exogenously added CysRsaA incorporates in a manner favoring the poles and division plane (Figure 1h). However, reconstituting S-layer assembly in this way creates more widespread, punctate fluorescence signal on the cell body (Figure 1h,j, blue) compared to the crack-like features seen in native S-layer assembly (Figure 1f,j, red). This effect is attributable to structural differences between the native and exogenous S-layers and is discussed below. Previous immuno-gold staining and electron microscopy of RsaF, the outermost component of the RsaA secretion apparatus, indicated its diffuse localization^{23,26}. Therefore, the non-uniform reconstitution of S-layer assembly with purified protein suggests that a factor independent of RsaA secretion contributes to assembly localization.”

Native S-layer assembly features on the cell body appear crack-like while the exogenous case exhibits more widespread, small punctate features, indicating structural

differences between the two. The reviewer's hypothesis regarding differences in crack generation/filling between native and exogenous S-layer assembly is supported by Figure 3, which shows large native S-layer crystals, but small ones in the reconstituted case. This finding is further exemplified in Supp Fig. 7, where adding sub-saturating amounts of exogenous CysRsaA to Δ RsaA cells shows many small puncta. Therefore, the change in morphology from longer crack-like features observed in native assembly to more widespread, punctate features in the exogenous case reflects boundaries created by large versus small S-layer protein crystals. The many small crystals that make up the exogenously created S-layer will necessarily present more boundaries for RsaA incorporation, leading to increased and less-continuous signal on the cell body. **The discussion section of the manuscript has been modified to consider these morphological differences.**

Discussion: Lines 216-229

“Our results imply that continuous **native** S-layer crystallization occurs at gaps, defects, and grain boundaries within the S-layer structure caused by localized cell wall growth or the inherent topology of the cell surface (Figure 4g). **Reconstituting S-layer assembly using the stepwise addition of exogenously purified CysRsaA yielded a non-uniform pattern of S-layer assembly, indicating a secretion-independent process (Figure 1j). During this experiment, the long crack-like features observed on the cell body during native assembly (Figure 1f) are replaced by more widespread, punctate fluorescence signal (Figure 1h). Reconstituting the S-layer with purified CysRsaA creates small, punctate S-layer crystal patches on the cell surface (Figure 3g, Supp Fig. 7), which are significantly smaller than natively produced S-layer crystals (Figure 3c,g). Therefore, the morphological difference in fluorescence signal on the cell body between native and exogenous cases could reflect the difference in molecular boundaries created by large versus small S-layer crystals. Small crystals present more boundaries for new S-layer incorporation, rationalizing the increased and less localized fluorescence signal along the cell body observed when reconstituting S-layer assembly by exogenous protein addition.**”

While the authors acknowledge that *in situ* AFM may resolve the molecular boundaries created by S-layer protein crystals, successful application of this technique would require single-digit nanometer resolution over a large area on a curved, living sample. We feel that the images and analysis presented in this study are sufficient to describe and rationalize the nano-scale features considered by the reviewer and that a study of S-layer assembly by *in situ* AFM is beyond the scope of this work.

Reviewer #2 Remarks

Comerci, C., et al. use super-resolution microscopy techniques to follow the diffusion and assembly of the Caulobacter crescentus S-Layer protein on the bacterial outer membrane. To date, self-assembly of the S-layer protein, RsaA, have been primarily studied by molecular biology or in vitro studies. RsaA along with other S-layer proteins from different bacterial species have been extensively studied using high resolution imaging techniques such as AFM and cryoEM. This manuscript offers a complete study of the secretion and self-assembly mechanism of RsaA on a bacterial cell surface. Using STED microscopy and single molecule tracking, the authors learn that the topology of this crescent shaped bacterial influences the self-assembly of RsaA. Sub-diffraction microscopy techniques were key to imaging these assembly dynamics and elucidating the mechanism. The experiments and controls in this manuscript are well conceived. The scientific conclusions that are drawn are sound and reveal previously unknown aspects of the S-layer self-assembly process in vivo. I invite the authors to revise the manuscript by addressing the following comments.

The authors thank the reviewer for their close reading and constructive comments.

Reviewer #2 Specific Comments

The p4A plasmid system in the C. crescentus strain JS1023 was used in this manuscript and shown to express the same levels of RsaA as well as the CysRsaA. Recent work on RsaA expression and display has reported that the p4B plasmid may result in over-expression of RsaA as well as plasmid instability (Charrier et al 2019, DOI: 10.1021/acssynbio.8b00448). Does the expression rate of RsaA affect secretion and assembly of this S-Layer on the surface?

This is a useful point and we are happy to clarify. We observed no issues with plasmid instability despite culturing each strain in liquid for three days before each experiment. Strains were inoculated into rich media on the first day, transferred to minimal media on the second, and then imaged on the third. Complete fluorescent labeling of the assembled S-layer indicates stability of the plasmid containing CysRsaA. Furthermore, we have updated the methods section to note the use of 2 µg/mL Chloramphenicol in liquid C. crescentus cultures to continually select for cells bearing the p4A plasmid encoding CysRsaA. The authors note that the location of the mutation used in this study is several hundred amino acids upstream of the edited sites in the referenced manuscript, which could contribute to the observed differences in expression characteristics.

Lines 293-294

“All liquid and plated cultures involving CysRsaA included 2 µg/mL chloramphenicol.”

The RsaA expression gel shown in Supp Fig. 1 displays only soluble protein that could be extracted from the cell surface, indicating that the production of a correctly assembled S-layer is occurring at the same rate as in WT cells. Additionally, previous extensive electron microscopy of the CysRsaA (JS1023) strain used in this study showed an intact, assembled S-layer covering the cell surface (Amat et al 2010, DOI: 10.1128/JB.00747-10). Therefore, we reason that the rate of secretion of this strain is similar to that of WT and does not affect its assembly on the surface.

Would stably integrating the CysRsaA gene in the genome with the native promoter affect the dynamics of assembly?

A worthwhile goal, but attempts to stably integrate a mutant RsaA sequence into the genome led to sub-saturating quantities of S-layer protein (i.e. cells did not produce RsaA fast enough to cover the cell entirely during log phase growth). “Over” expression via the p4A plasmid along with the repBAC plasmid replication system was our solution to bring CysRsaA expression up to native quantities, which we could not achieve by stable genome integration.

On page 5, line 90 the manuscript states that “Saturating quantities (600 nM) of DY-480XL-labeled CysRsaA were introduced...” How was this RsaA protein concentration determined to be saturating? Was the saturation determined relative to the total surface area of all the cells in solution?

A calculation of the amount of protein required to cover a culture of 2×10^8 cells/mL ($OD_{600nm} = 0.5$ as measured by colony forming units) with a surface area of $8.5 \mu m^2$ per cell yields a concentration of 45 nM RsaA. Supp Fig. 7 shows sub-saturation at 200 nM, indicating that not all added protein sticks to the cells. Therefore, protein saturation was determined empirically, and we found that 600 nM CysRsaA completely covers Δ RsaA cells. **A representative single-color control image from this experiment has been added as Supp Fig. 2d (below).** A higher amount was not pursued due to the material demands of the experiment, which required micromolar concentrations of purified, labeled CysRsaA protein. The concentration of 600 nM CysRsaA is therefore theoretically and empirically sufficient to saturate S-layer binding sites on the cellular surface.

On page 5, line 93 the manuscript states that exogenously added DY-480XL-labeled CysRsaA incorporates preferentially at the poles and division plane in addition to increased incorporation along the body. However, Figure 1j indicates that the fraction of labeled CysRsaA incorporation along the body is similar to that at the poles. Later in Figure 3f,g,j you again show that exogenously added labeled CysRsaA results in even incorporation of the protein along the cell. The explanation of these data sets should be more consistent and specific to avoid ambiguity.

We agree that this point needs clarification, and we are happy to do so. During exogenous addition (Figure 1h,j blue), the cells were given 30 minutes to continue growth before the second round of labeling, allowing for S-layer subunit incorporation and assembly. This experiment showed that even though the magnitude of the signal at mid-cell is similar to the signal at the poles, this signal is non-uniform and exhibits notable dips at the non-polar regions. This indicates that a factor not attributable to secretion exerts some amount of localization control over S-layer assembly during normal cell growth. On the other hand, the exogenous experiments shown in Figure 3f,g,j are single-color experiments imaged without additional cell growth. In this case (Figure 3j, blue), we observe a more uniform signal compared to the case of exogenous growth (Figure 1j, blue), especially when considering the non-polar regions. **The manuscript text has been modified to clarify this aspect of the study.**

Results: Lines 92-103

“This experiment revealed that exogenously added CysRsaA **incorporates in a manner favoring the poles and division plane (Figure 1h). However, reconstituting S-layer assembly in this way creates more widespread, punctate fluorescence signal on the cell body (Figure 1h,j, blue) compared to the crack-like features seen in native S-layer assembly (Figure 1f,j, red). This effect is attributable to structural differences between the native and exogenous S-layers and is discussed below.** Previous immuno-gold staining and electron microscopy of RsaF, the outermost component of the

RsaA secretion apparatus, indicated its diffuse localization^{23,26}. Therefore, the non-uniform reconstitution of S-layer assembly with purified protein suggests that a factor independent of RsaA secretion contributes to assembly localization.”

Discussion: Lines 216-229

“Our results imply that continuous native S-layer crystallization occurs at gaps, defects, and grain boundaries within the S-layer structure caused by localized cell wall growth or the inherent topology of the cell surface (Figure 4g). Reconstituting S-layer assembly using the stepwise addition of exogenously purified CysRsaA yielded a non-uniform pattern of S-layer assembly, indicating a secretion-independent process (Figure 1j). During this experiment, the long crack-like features observed on the cell body during native assembly (Figure 1f) are replaced by more widespread, punctate fluorescence signal (Figure 1h). Reconstituting the S-layer with purified CysRsaA creates small, punctate S-layer crystal patches on the cell surface (Figure 3g, Supp Fig. 7), which are significantly smaller than natively produced S-layer crystals (Figure 3c,g). Therefore, the morphological difference in fluorescence signal on the cell body between native and exogenous cases could reflect the difference in molecular boundaries created by large versus small S-layer crystals. Small crystals present more boundaries for new S-layer incorporation, rationalizing the increased and less localized fluorescence signal along the cell body observed when reconstituting S-layer assembly by exogenous protein addition.”

On page 8, the paragraph beginning on line 8 states that calcium depletion results in shedding of the RsaA S-layer. The referenced article (Ref. 30) indeed shows RsaA sheds when first grown in M2G media with high calcium and then shifted to low (8 μ M) calcium. However, this shift to a low calcium media also results in increased aggregation of RsaA. At 50 μ M calcium, presumably the RsaA is being excreted and should attach to the cell LPS even though it is not crystalline. The authors should show whether aggregates are present in these cultures (Figure 3a shows only crystalline RsaA) and address whether aggregates may affect the kinetics of RsaA nucleation and assembly as observed in Figure 3c and 3e.

Decreasing the calcium concentration in the media to 50 μ M causes aggregation and shedding of the S-layer as the reviewer and Ref. 30 note. Fluorescent images of cells that have shed their S-layer (Figure 3c, $t = 0$) show no residual fluorescent labeling indicating either a failure to label the aggregated protein or that the protein is suspended in the media. As the reviewer hypothesizes, stationary phase cultures of CysRsaA or WT cells grown at low calcium concentrations show evidence of macroscopic protein aggregation

(see image below). Dissolution of these aggregates and subsequent analysis by SDS-PAGE reveal their identity to be predominantly the S-layer protein (see large band in gel below). Therefore, we reason that the S-layer is completely and cleanly shed upon overnight calcium depletion and that suspended RsaA aggregates should not affect the nucleation and assembly patterns observed in Figures 3c and 3e.

In Figure 3g,k, the authors add exogenous STAR Red-CysRsaA and measure puncta that form on the cell surface. These puncta are determined to have ~50 monomers of RsaA using photon counting. Is the invariance in size of the puncta with increasing RsaA concentrations due to the incubation time?

In this experiment (Figure 3g,k, both updated in this revision), very low (sub-saturating) concentrations of protein are added to cell cultures while the incubation time is held constant. Performing two-color addition of CysRsaA protein to these cells (Figure 3i) showed growth of existing puncta as well as the formation of new ones, indicating that the nucleation process is fast in comparison to the 15 min incubation time. Therefore, it is unlikely that incubation time affects the formation of the observed puncta. Rather, the concentration-dependence of the number of puncta combined with the invariance in size (measured by photon counting) suggest that the minimum size of the puncta are determined by the size of the critical nucleus of an S-layer crystal, as the reviewer describes below.

The manuscript mentions in the Materials and Methods section that incubation is typically 15 minutes. If nucleation occurs faster than the incubation time, the puncta reflect nucleation

followed by some growth. Moreover, the number of spots/cell appear to increase in an exponential fashion with the concentration of CysRsaA. Fitting a simple nucleation model may give a minimal nucleus size for RsaA on the cell surface.

We agree that nucleation is faster than incubation time, as evidenced by the two-color experiment (Figure 3i) and that the puncta in Figure 3g,k reflect the nucleation of protein crystals. We have repeated the experiment in Figure 3k with lower-density cultures ($OD_{600nm}=0.2$) and now provide 11 data points instead of 4 to provide a more precise estimate of the size of the critical nucleus for RsaA crystals on the cell surface. The lower cell density shifts the window of measurable concentrations since the small size of the bacterium limits the maximum number of individual crystals that can be resolved by STED microscopy. In the updated data set, we observe a roughly linear increase in nucleation sites but a relatively constant number of molecules per crystal, indicating that crystal nucleation is fast even at low concentrations. This near-linear increase in number of protein crystals indicates that crystal growth is not measurably contributing to the size of the crystals, and instead nucleation is the dominant phenomenon in this experiment. We predict that nucleation should give way to crystal growth at higher concentrations, in agreement with the reviewer’s suggestion to fit an exponential model. However, since near-linear nucleation signal suggests crystal growth is not at play during the 15 min incubation, we can average the number of molecules per spot from concentrations 2-10 nM to yield a robust estimate of 49.3 molecules required to form a critical crystal nucleus of RsaA. Figure panels 3g and 3k and associated textual references have been modified to reflect this more complete dataset.

Results Lines 152-165:

“To further evaluate this mode of assembly, very low concentrations (0.5 to 10 nM) of purified, STAR RED labeled CysRsaA were added to cultures of Δ RsaA cells ($OD_{600nm}=0.2$) and puncta were observed (Figure

3f,g). The number of puncta appeared dependent on CysRsaA concentration whereas the number of molecules in each punctum did not **strongly** correlate with CysRsaA concentration from **0.5 to 10** nM (Figure 3g,k). **These observations are consistent with fast nucleation occurring at the measured CysRsaA concentrations and limited further growth of the observed puncta during the 15 min incubation.** Adding 5 nM of exogenously purified and labeled DY-480XL CysRsaA...”

Discussion Lines **252-254:**

“Remarkably small but stable RsaA protein crystals consist of only ~50 molecules (Figure 3g,k), suggesting that **fast**, efficient nucleation at low concentrations may be another selectable trait supporting protein self-assembly.”

The authors show S-layer growth data in Figure 3e,i. Is it possible to estimate the observed growth of RsaA S-layers on the cell surfaces? Can an estimate of the growth rate be determined and compared to crystal growth rates obtained for other S-layers?

We have analyzed the time-resolved *de novo* S-layer growth rate (Figure 3c) by plotting the surface area coverage as a function of time after calcium addition, **which has been added as Supp Fig. 6c (below)**. Due to the inherent errors in this type of experiment and the 3D nature of the bacterial surface, a precise estimate for S-layer growth rate cannot be extracted from this data set. However, utilizing the 94% coverage we observe after 120 mins of *de novo* S-layer growth, we can roughly estimate a rate of 0.8% surface area coverage/min. Assuming an average bacterial surface area of 8.5 μm^2 , this leads to an S-layer growth rate of roughly 0.07 $\mu\text{m}^2/\text{min}$. The concentration dependence of SLP crystallization (Herrmann et al., Biophys J 2017) renders a comparison to other S-layer proteins, typically studied *in vitro*, of limited utility.

In the SMT experiments, the manuscript does not explain why two different dyes other than DY-480XL and Star Red were used? Were the AlexaFluor dyes more stable? Are the AlexaFluor dyes also cell impermeable?

As is usual, the fluorescent dyes were chosen according to the needs of the different experiments. DY-480XL and STAR RED are fluorophores specifically optimized for STimulated Emission Depletion (STED) microscopy. Single-molecule tracking was performed on a wide-field microscope rather than a confocal/STED microscope (Supp Fig. 10). Cell-impermeable Sulfo-Cy3 was chosen as a diffraction-limited indicator for S-layer seed crystals. To obtain single-molecule tracks of several minutes in duration, a particularly bright and photostable single-molecule fluorophore was required, and we selected cell-impermeable AlexaFluor647 for the SMT measurements. **The methods section has been modified to reflect these details.**

Methods: Lines **332-337**

“Overnight labeling of CysRsaA protein (>20 μ M) was performed on ice with the addition of 1 mM TCEP and at least 5-fold stoichiometric excess of maleimide-derivatized, **cell-impermeable fluorescent dyes, STAR RED and DY-480XL, which are well-suited for confocal and STED imaging. Cell-impermeable Sulfo-Cy3 (Lumiprobe) was used as a diffraction-limited indicator for S-layer seed crystals. AlexaFluor647 (ThermoFisher) was used to track single CysRsaA molecules since SMT requires a particularly bright and photostable fluorophore.**”

In Supplemental Figure 9, it is not clear how the molecules in each panel were classified. Which molecule was classified to be bound, bound and close to a nucleation site, or close to a

nucleation site? In addition, a color bar indicating the time for the Upper panels in each figure would be helpful.

The authors thank the reviewer for pointing out this issue. The classification criteria were previously described in the Methods section, noted in the figure caption of Figure 4c, and supported by Supp Fig 8. We now provide vertical, colored shading to Supp Fig. 9 consistent with the colors in the Venn diagram in Figure 4d to clarify the state of the molecule at any given time point. Supp Fig. 9a,b is reproduced below as demonstration.

On page 10, line 157 SMT is introduced without defining the acronym “Single Molecule Tracking.”

The text has been modified to define single-molecule tracking.

Lines 181-183

“Therefore, we employed single-molecule tracking (SMT) to dynamically track the location of individual CysRsaA monomers anchored to the LPS outer membrane.”

Reviewer #3 Remarks

The manuscript by Comerci, Herrmann, et al. investigates the assembly of the surface layer in C. crescentus bacteria using STED microscopy and single-molecule tracking. Based on their observations, they propose a model of S-layer assembly whereby RsaA monomers are secreted diffusely on the outer membrane and diffuse until incorporated into S-layer crystals where they contribute to the 2D crystal growth.

The manuscript is well-written, experiments are conducted carefully using a range of cutting-edge imaging methods and the results are convincing. The findings and model are of significant interest in that they shed light onto this fundamental mechanism of self-assembly which might also have an impact in future drug development. I therefore support publication in Nature Communications.

We thank the reviewer for their consideration and constructive comments.

Reviewer #3 Specific Comments

(1) In Fig. 1, the native RsaA seems to form a smoother distribution (Fig. 1f) than the exogenously added RsaA which appears in more puncta-like structures (Fig. 1h). The images in Supp Fig.3 show a similar pattern. It would be interesting to hear a possible explanation for this phenomenon from the authors, if it should turn out to be more than just a coincidence from the selected images.

We agree that native S-layer assembly features on the cell body appear crack-like while the exogenous case exhibits more widespread, punctate features, indicating structural differences between the two. We attribute these differences to a change in the type of boundaries created by native or exogenously created S-layer crystals. This is evidenced in Figure 3, which shows large native S-layer crystals, but small reconstituted ones. This finding is further exemplified in Supp Fig. 7, where adding sub-saturating amounts of exogenous CysRsaA to Δ RsaA cells shows many small puncta. Therefore, the change in morphology from longer crack-like features observed in native assembly to more widespread, punctate features in the exogenous case reflects boundaries created by large versus small S-layer protein crystals. The many small crystals that make up the exogenously created S-layer will necessarily present more boundaries for RsaA incorporation, leading to increased and less-continuous fluorescence signal on the cell body. The manuscript text and discussion have been modified to clarify these findings.

Results: Lines 92-103

“This experiment revealed that exogenously added CysRsaA incorporates in a manner favoring the poles and division plane (Figure 1h). However, reconstituting S-layer assembly in this way creates more

widespread, punctate fluorescence signal on the cell body (Figure 1h,j, blue) compared to the crack-like features seen in native S-layer assembly (Figure 1f,j, red). This effect is attributable to structural differences between the native and exogenous S-layers and is discussed below. Previous immuno-gold staining and electron microscopy of RsaF, the outermost component of the RsaA secretion apparatus, indicated its diffuse localization^{23,26}. Therefore, the non-uniform reconstitution of S-layer assembly with purified protein suggests that a factor independent of RsaA secretion contributes to assembly localization.”

Discussion: Lines 216-229

“Our results imply that continuous native S-layer crystallization occurs at gaps, defects, and grain boundaries within the S-layer structure caused by localized cell wall growth or the inherent topology of the cell surface (Figure 4g). Reconstituting S-layer assembly using the stepwise addition of exogenously purified CysRsaA yielded a non-uniform pattern of S-layer assembly, indicating a secretion-independent process (Figure 1j). During this experiment, the long crack-like features observed on the cell body during native assembly (Figure 1f) are replaced by more widespread, punctate fluorescence signal (Figure 1h). Reconstituting the S-layer with purified CysRsaA creates small, punctate S-layer crystal patches on the cell surface (Figure 3g, Supp Fig. 7), which are significantly smaller than natively produced S-layer crystals (Figure 3c,g). Therefore, the morphological difference in fluorescence signal on the cell body between native and exogenous cases could reflect the difference in molecular boundaries created by large versus small S-layer crystals. Small crystals present more boundaries for new S-layer incorporation, rationalizing the increased and less localized fluorescence signal along the cell body observed when reconstituting S-layer assembly by exogenous protein addition.”

(2) The authors did a good job analyzing and presenting the results of the STED images (Supp Fig. 3). I understand the focus of this paper is about the new RsaA signals. However, since the authors already analyzed the old RsaA signals, it would be interesting to show the curves of old signals as well (Fig. 1j).

We found that old S-layer signal is largely complementary to new S-layer signal and have added analysis of old S-layer signal to Supp Fig. 3a, point 7 as demonstration.

7. Calculate fraction of cells with S-layer

(3) The statement in lines 112-114 “If PG insertion were the only driver of S-layer assembly, disruption should have occurred only at the stalked pole, rather than both.” is hard to follow without additional explanations.

We thank the reviewer for pointing out this issue. The sentence has been replaced to clarify this statement.

Lines 116-119

“At 25 $\mu\text{g/mL}$ A22, bipolar localized S-layer assembly is disrupted as evidenced by a decrease in signal at both poles (Figure 2b,c, black). Native cell wall growth is asymmetric to maintain the shape of the cell and lengthen the stalk^{1,2}, yet S-layer assembly remains largely symmetrically localized at both poles with or without A22.”

(4) It would be helpful to report the approximate resolution of the STED images in both channels to put the observations in context of the size of individual RsaA molecules, etc.

We thank the reviewer for catching this oversight. The methods section has been modified to reflect the resolution of the STED channels.

Lines 371-372

“STED images have a FWHM resolution of ~ 60 nm and ~ 80 nm in the red and green channels, respectively^{22,47}.”

Reviewer #4 Remarks

*In this work, the authors used super-resolution fluorescence microscopy and a pulse-chase scheme to propose that localised S-layer assembly in *C. crescentus* is driven by areas of high Gaussian curvature, and that S-layer growth relies on self-assembly of RsaA molecules. The authors propose a mechanism for S-layer assembly that involves diffusion of secreted RsaA molecules into the growing S-layer and incorporation at compromised sites. This work builds on previous in vitro studies showing RsaA self-assembly, and would be of interest to people in their specific field. Given that the S-layer integrity is important for cell viability do I find it curious and maybe even unlikely that its assembly would be purely diffusion driven. Overall, the paper is well-written for the most part, experiments are sound, and the data carefully interpreted. However, regrettably I don't see this study attract attention from the large and diverse readership of Nature Communications.*

In *C. crescentus*, S-layer integrity is not required for cell viability under laboratory conditions as evidenced by the genomic knockout strain used in this study, Δ RsaA. However, S-layer integrity has been linked to several important functions such as protection from predators (Fuente-Nunez et al., Appl Environ Microbiol 2012) and nutrient acquisition (Herrmann et al., Biophys J 2017). Diffusion-driven processes are common in micrometer-sized bacteria, as no motors are present. To further discuss the plausibility of a diffusion-driven model, the text has been modified to consider a possible molecular basis for monomer diffusion on the cellular surface. Crystal structures of the S-layer anchoring domains from the gram-positive bacteria *Paenabacillus alvei* and *Bacillus anthracis* reveal multiple polysaccharide binding sites per domain. It is therefore plausible that S-layer proteins might migrate along the cell surface using their multiple polysaccharide binding sites by alternately binding and releasing the polysaccharides of neighboring LPS molecules.

Discussion Lines 204-215

“While SMT of RsaA was performed on Δ RsaA cells, it is plausible that RsaA diffuses either on or within the LPS outer membrane, depending on whether an existing S-layer structure is present above the site of RsaA secretion. Protein diffusion within the LPS outer membrane in gram-negative bacteria has been observed by fluorescence recovery after photobleaching, but individual LPS molecules are immobile³³. In *C. crescentus*, the LPS outer membrane is ~18 nm thick, while the size of a single RsaA monomer can be approximated by its radius of gyration ($R_g=5.8$ nm)^{14,30}. Several crystal structures of S-layer anchoring domains from gram-positive bacteria have been elucidated, revealing multiple polysaccharide binding sites per subunit^{34,35}. Assuming that the N-terminal domain of RsaA has similar multiple carbohydrate binding surfaces, a plausible molecular

basis for anchored extracellular protein diffusion is that S-layer proteins migrate along the cell surface by alternately binding and releasing the polysaccharides of neighboring LPS molecules.”

The findings described in this study are of interest to several audiences. First, S-layers are involved in pathogenesis for multiple common pathogens. The self-assembly pathway elucidated here holds implications for antibiotic design strategies targeting the S-layer and the bacterial surface in general. Second, SLPs are currently used and being developed for a variety of nanotechnological and biotechnological applications. Knowledge and control of S-layer assembly in native and reconstituted environments will assist in further engineering these unique nanomaterials. Third, S-layers are found on almost all archaea and many different types of bacteria. Unveiling the S-layer assembly pathway in *C. crescentus* allows insight into the evolutionary basis for the construction of these ubiquitous macromolecular structures. The authors feel that these implications are reasonably discussed in the existing manuscript text.

Reviewer #4 Specific Comments

No major criticisms, but some suggestions that could be considered to strengthen their claims:

Line 121 and Figure 2: The authors suggest that, in A22-treated cells, increased incorporation of CysRsaA along the cell body is due to delocalized PG insertion. Apart from the graphical depiction in Fig. 2b, the authors do not show the effect of A22 on sites of PG insertion. It would be more convincing if sites of PG insertion could be labelled (eg. using FDAAAs) and correlated with sites of S-layer assembly.

We agree that imaging PG insertion would be appealing, but we already performed a number of control experiments to show that A22 treatment in our hands matched the results of previous publications addressing cell wall growth. To summarize, pulse-chase electron microscopy of the *C. crescentus* cell wall sacculus revealed highly localized sites of PG insertion at the base of the stalk and along the middle of the cell (Aaron et al., Mol Micro 2007). These locations are consistent with lengthening the cell body and stalk during the cell cycle (Figge et al., Mol Micro 2004). Previous fluorescence microscopy revealed that PG insertion machinery in *C. crescentus* is dynamically localized in helical structures focused at the middle of the cell in order to maintain the cell's crescent shape (Dye et al., PNAS 2005; Dye et al., Mol Micro 2011). However, this localized activity is highly manipulable using A22, a cell-permeable, specific, fast-acting small-molecule inhibitor (Gitai et al., Cell 2005). To corroborate this previous work and validate that the cells and drugs behaved predictably in our hands, we performed growth curves and phase contrast imaging of WT and CysRsaA cells treated with A22 (Supp Fig. 4). Our dose-dependent findings were in good agreement with reports from the literature for both population

growth and morphological phenotypes (Gitai et al., Cell 2005; Takacs et al., J Bacteriol 2010). Low dose A22 (2 µg/mL) caused lemon-shaped cells to form, indicating diffuse PG insertion, while high dose (25 µg/mL) stopped growth entirely (Supp Fig. 4).

Altogether, previous work shows that the asymmetric shape of *C. crescentus* necessitates asymmetric cell wall production, which provides a reasonable basis for the graphical PG representation in Figure 2b. We agree that an imaging experiment labeling both the cell wall and S-layer assembly would be reasonable; however, this experiment would be technically difficult to achieve given the current lack of a PG label capable of STED imaging. We reasoned that technical feasibility concerns combined with the strong knowledge basis for this experiment did not require correlative PG and S-layer assembly imaging. To clarify, the text has been modified to describe the relationship between the experiment in Figure 2 and asymmetric cell wall growth in *C. crescentus*.

Results: Lines 116-119

“At 25 µg/mL A22, bipolar localized S-layer assembly is disrupted as evidenced by a decrease in signal at both poles (Figure 2b,c, black). Native cell wall growth is asymmetric to maintain the shape of the cell and lengthen the stalk^{1,2}, yet S-layer assembly remains largely symmetrically localized at both poles with or without A22.”

Discussion: Lines 230-233

Increasing localized cell wall growth along the cell body (treatment with 2 µg/mL A22) increases S-layer assembly at that location; however, preventing asymmetric cell elongation (treatment with 25 µg/mL A22) disrupts S-layer assembly at both poles, indicating that an additional factor is coordinating this process (Figure 2c)².

Concerning Figure 3a (line 125 onwards), it is not immediately clear that the media is exchanged, the word “reintroducing” can be changed to avoid confusion.

The manuscript text has been modified to clarify that for the *de novo* assembly experiment (Figure 3a), the concentration of calcium in the media was adjusted from 50 µM to 500 µM without media exchange.

Results: Lines 137-141

“Calcium depletion (50 µM CaCl₂ instead of 500 µM CaCl₂ normally present in minimal growth medium) has been shown to cause shedding of the RsaA S-layer^{29,30}. After culturing CysRsaA cells overnight in media containing 50 µM CaCl₂, supplementing the media with 500 µM CaCl₂ causes a new S-layer to be produced on the surface of *C. crescentus* (Figure 3a).”

Lines 95-98: The authors claim that diffuse localisation of the RsaF component of the RsaA secretion machinery and incorporation of RsaA to the cell poles and division plane suggests that S-layer assembly is independent of RsaA secretion. Perhaps the authors could correlate RsaF localisation with S-layer assembly in the same cell. However, the authors also infer diffusion of RsaA molecules from particle tracking experiments (Fig. 4b), which also suggests that S-layer assembly is independent of RsaA secretion, so this is not so critical to show.

The authors agree with the reviewer’s reasoning and therefore do not consider it prudent or necessary to perform further imaging experiments of the RsaA secretion machinery. Even so, this section has been modified for clarity.

Results: Lines 100-103

“Previous immuno-gold staining and electron microscopy of RsaF, the outermost component of the RsaA secretion apparatus, indicated its diffuse localization^{23,26}. Therefore, the non-uniform reconstitution of S-layer assembly with purified protein suggests that a factor independent of RsaA secretion contributes to assembly localization.”

In its current form does this manuscript read like it has been written up as a short communication. The figures are therefore sometimes very dense and is consequently quite hard to follow. Perhaps would it be advisable to split the figure to make them more reader friendly?

The authors thank the reviewer for their close reading of the manuscript. We chose to group results together based on logical connection.

Reviewers' Comments:

Reviewer #2:

Remarks to the Author:

The authors of this manuscript have addressed the comments by myself and the other reviewers through additional data, figures, and edits to the manuscript. I believe the points raised by the reviewers have been satisfactorily addressed.

Reviewer #3:

Remarks to the Author:

The authors have addressed all my comments extensively and to my full satisfaction. This is a very strong manuscript, in my opinion and I strongly support publication of the manuscript in Nature Communications.